# Belief inference for hierarchical hidden states in spatial navigation
Risa Katayama [1,2] ✉, Ryo Shiraki[1], Shin Ishii[1,3,4] & Wako Yoshida[5,6]

Uncertainty abounds in the real world, and in environments with multiple layers of unobservable hidden states, decision-making requires resolving uncertainties based on mutual inference. Focusing on a spatial navigation problem, we develop a Tiger maze task that involved simultaneously inferring the local hidden state and the global hidden state from probabilistically uncertain observation. We adopt a Bayesian computational approach by proposing a hierarchical inference model. Applying this to human task behaviour, alongside functional magnetic resonance brain imaging, allows us to separate the neural correlates associated with reinforcement and reassessment of belief in hidden states. The imaging results also suggest that different layers of uncertainty differentially involve the basal ganglia and dorsomedial prefrontal cortex, and that the regions responsible are organised along the rostral axis of these areas according to the type of inference and the level of abstraction of the hidden state, i.e. higher-order state inference involves more anterior parts.

In everyday behaviour, the current state of an environment often remains hidden from direct observation and must be inferred through belief. These beliefs concerning hidden states inherently carry uncertainties stemming from two distinct sources. First, the hidden state may not be inferred accurately because only insufficient information is available or the information is inaccurate, i.e., due to observational uncertainty. The second source of uncertainty arises when the hidden state cannot be uniquely identified (i.e., two or more states look exactly the same, even with perfect observation). This uncertainty occurs due to the inherent structure of the environment itself, i.e., state uncertainty.

This complex scenario, involving the simultaneous handling of these different levels of hidden states, is often manifest in real-world spatial navigation challenges. For example, consider navigating the Gion Festival in Kyoto, which is known for its beauty and bustling crowds. The streets form a grid pattern resembling a Go board and successfully navigating through the festival requires one to identify your current location, whilst avoiding several floats, the centrepieces of the festival, where you can easily get stuck in the crowd. The floats are strategically positioned throughout the city, and a distinctive musical accompaniment known as the Gion Bayashi can be heard as you approach them. The direction from which this sound is heard serves as a vital clue for identifying one's location. However, the auditory information contains uncertainty as the floats can move around. In such

scenarios, both the perceived sound direction and one's location in the city constitute hidden states that need to be estimated. This decision-making process, guided by the estimation of these hidden states, can be formulated as a partially observable Markov decision process (POMDP). In cases featuring a hierarchy of hidden states, this problem can be classified as a hierarchical POMDP. Despite the success of hierarchical POMDP models as machine learning methods and their widespread use in artificial intelligence for facilitating adaptive behaviour[1–6], our understanding of how the brain processes uncertain information to solve such challenges remains limited.

In this study, we computationally examined inference processes in the human brain and investigated how various brain regions contribute to the inference of multi-layered hidden states. In neuroscience, hierarchical structures have proven useful as neural and behavioural models for explaining complex perceptual learning[7–12], motor learning[13–15], and social decision-making[16,17]. Neuroimaging investigations focused on prediction problems within hierarchical state spaces have revealed that predictions at different levels of hierarchy correlate with neural activity in distinct brain regions, particularly in perceptual[12] and reward-based decision-making[18]. We posit that this segmentation of neural activity based on hierarchy also extends to higher cognitive systems. Several brain imaging studies have examined spatial navigation tasks involving uncertainty[19–21] and have identified activity in the medial prefrontal cortex (mPFC) with respect to

[1]Graduate School of Informatics, Kyoto University, Kyoto 606-8501, Japan. [2]Department of AI-Brain Integration, Advanced Telecommunications Research Institute International, Kyoto 619-0288, Japan. [3]Neural Information Analysis Laboratories, Advanced Telecommunications Research Institute International, Kyoto 619-0288, Japan. [4]International Research Center for Neurointelligence, the University of Tokyo, Tokyo 113-0033, Japan. [5]Department of Neural Computation for Decision-Making, Advanced Telecommunications Research Institute International, Kyoto 619-0288, Japan. [6]Nuffield Department of Clinical Neurosciences, University of Oxford, Oxford OX3 9DU, UK. ✉e-mail: katayama.risa.8d@kyoto-u.ac.jp

state inference[19]. The medial surface of the frontal region has been linked to beliefs about hidden states in studies on both non-human primates[22] and humans[23–25], leading to the hypothesis that different areas of this region may be involved in inferring different levels of hidden states within a hierarchical environment.

To test this hypothesis, we designed a Tiger maze navigation task that combines two distinct yet representative types of POMDP problems: the classic Tiger problem with observational uncertainty and partially observable maze navigation involving state uncertainty. In the partially observable maze problem, an agent estimates the current location within a maze whose structure has already been learned from scenes that show only a limited area around the current location[19]. Within this maze, multiple locations offer the same visual scene, establishing a one-to-many relationship between observations and hidden states. As the agent navigates through the maze, the increasing history of observations aids in identifying the hidden state by progressively narrowing down the possible locations. Here, we introduce the probabilistic observations used in the Tiger problem in this maze task. In the Tiger problem, an agent faces two doors, one of which conceals a tiger (the tiger door). The agent has the choice of opening either door or choosing to listen at the door for the tiger's roar (at a small cost), which can be observed probabilistically[26]. In the Tiger maze navigation task, each grid has four doors, including a tiger door, with the visual scene being identical in each grid, and listening actions can be performed as many times as desired. In other words, the grid location can only be inferred by inferring the position of the tigers, which have learned in the training session (see the "Methods" section, specifically the Training task). This arrangement provides probabilistic observations that can be used to estimate which door the tiger is likely to be behind, while simultaneously assisting in estimating the grid location in the maze. Thus, the Tiger maze navigation problem constitutes a POMDP with two distinct types of hidden states. In this problem, the position of the tiger door can be estimated directly from the observations, whereas the grid location can only be indirectly inferred using both these observations and map information. If the grid location can be identified, the position of the tiger door is uniquely determined, highlighting the hierarchical structure of this problem, with the grid location representing the higher level. Here, while most maze navigation studies have assumed an environment with an observable state and an explicit goal state and regarded the navigation process as a decision-making problem[27–29] or the generation of a spatial predictive map[30,31], the Tiger maze navigation task is a problem-setting focused on hidden-state inference.

## Results
### Behavioural results
In this study, twenty healthy participants engaged in the Tiger maze navigation task using a pre-trained maze (Fig. 1b). The goal of the game is to move around the maze without getting eaten by the tiger, until a termination condition is reached, based on either surviving for a set number of trials or visiting a set number of grids. The success rate, that is, the proportion of games where the participants completed their exploration without opening the tiger door, was $81.3 \pm 11.1\%$ (equivalent to $19.5 \pm 2.7$ out of 24 games in total). Each trial of the game begins with an action phase in which participants choose whether to move through the maze or listen to the tiger's roar, followed by a prediction phase in which they predict both the tiger door position and the grid location and report their confidence level for each (Fig. 1a). The mean prediction accuracy for the tiger door position and grid location was $80.6 \pm 3.5\%$ and $57.3 \pm 8.6\%$, respectively, and both prediction accuracies were significantly higher than chance (Wilcoxon signed-rank test; tiger door, $p = 8.9 \times 10^{-5}$; grid, $p = 8.9 \times 10^{-5}$). The prediction accuracies were also higher when the participants reported a high confidence level than when they reported a low confidence level (Fig. 1c, Wilcoxon signed-rank test; tiger door, $p = 8.9 \times 10^{-5}$; grid, $p = 8.9 \times 10^{-5}$).

During the action phase of the task, participants observed the scene and chose either to move or listen. It was impossible to infer the hidden current state from the scene alone, as the scene that could be observed was the same in all states. Thus, to correctly answer the predictions in the subsequent prediction phase, it was necessary to observe the direction of a tiger roar through listening action and infer the position of the hidden current grid and the tiger door. The participants indeed selected to listen more often in the early stages of maze exploration, and the probability of selecting to listen was negatively correlated with the number of trials per game (Fig. 2a, $r = -0.71$, $p = 3.5 \times 10^{-32}$). The proportion of listening actions also increased when the participants were less confident in their predictions (Fig. 2b, main effect for tiger door confidence, $F(1,72) = 84.5, p = 9.1 \times 10^{-14}$; main effect for grid confidence, $F(1,72) = 57.2, p = 1.0 \times 10^{-10}$; interaction, $F(1,72) = 4.1, p = 4.6 \times 10^{-2}$). Note that the significant decrease in the proportion of the Listen trials in the third trial is due to the fact that the participants typically selected their first move action in this specific trial index ($64.2 \pm 22.2\%$ of all games).

In the prediction phase, participants were required to predict both the tiger door position and the grid location in the maze. The tiger door inference is necessary to determine which doors can be safely opened in the current grid and can be estimated by listening to tiger roars. It can also be uniquely determined from the maze structure if the current grid location is known. However, inference of the grid is necessary for efficient exploration and can be estimated by gradually narrowing down the possible locations in the maze based on both the previous movement history obtained through multiple moving actions and the tiger door position predicted by listening actions. The two estimation processes are therefore bidirectional rather than independent; however, while the tiger door can be estimated directly from the observation obtained by choosing to listen, the grid can only be estimated indirectly from memory and movement history, which makes it a more challenging problem.

As the Listen and Move trial indices increased, confidence in predicting both the tiger door and the grid location increased, but with different trends. For the Listen trials, confidence in predicting the tiger door increased rapidly even early in the task (Fig. 2c left), and the correlation between the mean confidence level and the trial index was weak ($r = 0.35 \pm 0.43$), whereas for the grid location prediction, confidence increased gradually, i.e., the correlation was strong ($r = 0.77 \pm 0.27$); there was a significant difference in correlation ($p = 1.9 \times 10^{-3}$). On the other hand, when the Move was chosen, both the tiger door and grid predictions had a strong correlation between their confidence level and the trial index (tiger door: $r = 0.81 \pm 0.09$, grid: $r = 0.83 \pm 0.09$), with no significant difference in correlation ($p = 0.14$). The repeated two-way ANOVAs also showed significant effects on the tiger door confidence for both the trial index ($F(5,14) = 25.0, p = 1.6 \times 10^{-6}$) and the action type ($F(1,11) = 72.1, p = 3.7 \times 10^{-6}$), whereas for the grid confidence the effect of the trial index only was significant ($F(5,14) = 40.7, p = 7.5 \times 10^{-8}$; effect of the action type, $F(1,11) = 3.0, p = 0.11$). The prediction accuracies exhibited similar temporal profiles as the confidence levels for these two types of predictions (Fig. 2d; repeated two-way ANOVA, for the tiger door, effect of the trial index, $F(5,14) = 24.9, p = 1.7 \times 10^{-6}$; effect of the action type, $F(1,11) = 47.7, p = 2.6 \times 10^{-5}$; for the grid, effect of the trial index, $F(5,14) = 36.7, p = 1.5 \times 10^{-7}$; effect of the action type, $F(1,11) = 3.0, p = 0.11$). The difference in the temporal profiles of the confidence level evolution suggests that these two predictions involve different processes depending on the selected actions. Furthermore, the state transition frequency matrices show that the state transition from the low tiger door, low grid to the high tiger door, high grid confidence state was often mediated by the high tiger door, low grid confidence state (Fig. 2e–g). These results support the notion that tiger door inference was followed by the grid inference; in other words, the grid was hierarchically inferred after the tiger door inference in the Listen trials. The same statistical results were obtained when the data from the behavioural (outside of the scanner) and imaging experiments were analysed separately (Supplementary Fig. S1).

We implemented a hierarchical inference model in the brain that reflects behavioural results, in which the tiger door position and grid location in the maze are hierarchically inferred as probability distributions. This model assumed that when the participants performed the Listen action, they first inferred the tiger door position and then inferred the grid location. The

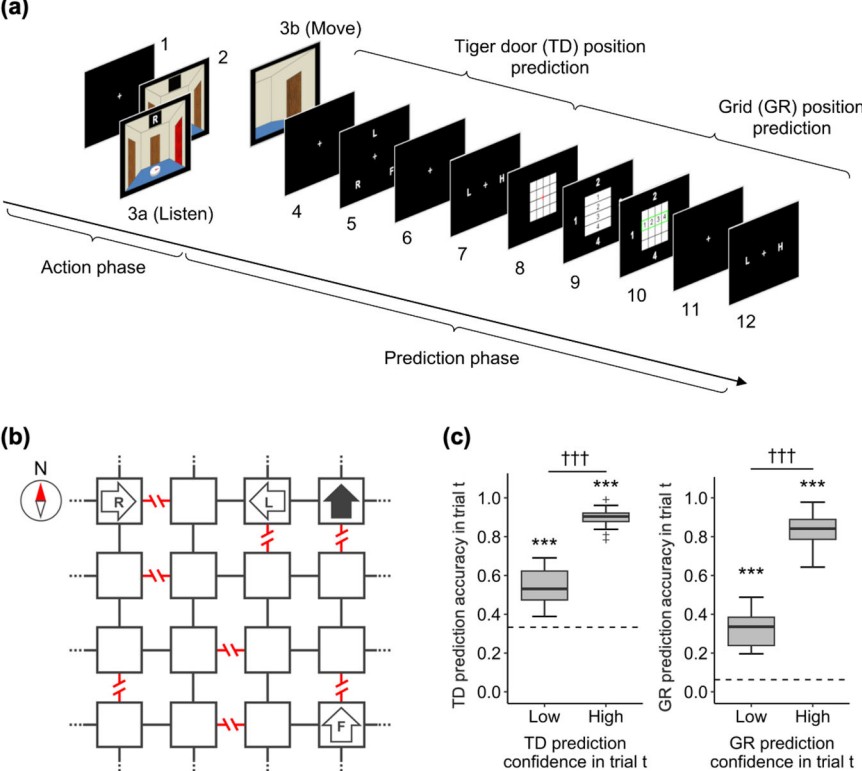

**Fig. 1 | Tiger maze navigation task.** Participants in the study navigated the pre-learned grid maze starting from an unknown initial state without opening the tiger doors, and were tasked with predicting both the position of the upcoming grid and the position of the tiger door in the upcoming grid. **a** Each trial comprised two phases: the action and prediction phases. In the action phase, participants observed the scene at their current state with the compass placed in the grid room (corresponding to step 2 in **a**). They then chose to move (left, forward, or right) to open the corresponding door and shift to an adjacent grid, or listen to detect a tiger roar from one of the three doors within a 4 s timeframe. If the participants chose to move, a short animation displayed the rotation of their body orientation and the opening of the door (step 3b, for 1.5 s). However, if the tiger door was chosen, the game concluded immediately without any score, and the participants proceeded to the next game. When the participants selected to listen, the door from which the tiger roar was heard turned red, and the location of the door was indicated using letters at the top of the screen (L, F, or R, indicating left, forward, and right, respectively), while the state remained unchanged. In the prediction phase, participants were first asked to predict the position of the tiger door in the upcoming state (step 4, 4 s) and report it by choosing one of three doors (L, F, or R; step 5, 2 s). They were then asked to evaluate their confidence level (step 6, 4 s) and indicate it by selecting either low or high (step 7, 2 s). Next, the participants were required to predict the upcoming grid location (step 8, 4 s) and report the coordinates by selecting one of four numbers,

each corresponding to the row and column coordinates on the maze (step 9 and 10, 2 s allotted for each). They then evaluated their confidence level regarding the grid prediction (step 11, 4 s) and reported it by selecting from the binary options (step 12, within 2 s). **b** Abstract layout of the maze. Each node represents a room in the maze, with grey edges representing normal (passable) doors, and red edges denoting tiger doors that should not be opened. Arrow symbols with the letters L, F, or R represent transitions in grid location and body orientation that result when a participant chooses to move left, forward, or right from the current state (dark grey arrow), respectively. Note that, the maze possessed a topologically torus structure, with the left and right columns and the top and bottom rows connected. For example, if a participant faced north in the first grid on the right of the top row and chose the front door, they would move to the first grid on the right of the bottom row, which also faced north. **c** The prediction accuracy significantly increased when the participants reported a high confidence level on the same trial for both the tiger door position prediction (left panel) and the grid location prediction (right panel) (Wilcoxon signed-rank test, †††$p < 0.001$). Each box extends from the lower to upper quartiles, with a horizontal line at the median. The whiskers indicate 1.5 × Interquartile range (IQR), with cross markers denoting outliers. The dashed lines signify chance levels, and asterisks indicate significance tested using a Wilcoxon signed-rank test compared with chance (***$p < 0.001$).

tiger door was inferred in a Bayesian fashion, i.e., it was updated as the product of the prior information, which is a prediction made in the previous trial, and the newly observed information (the position of the roaring door), weighted by an exponential parameter, delta. The estimated value of this delta parameter ($1.8 \pm 0.74$; Table 1) was greater than 1, indicating that the observed information was given more weight. The grid location was then also updated using Bayesian estimation, with the likelihood information being the extracted grid positions, such that the probability of satisfying the condition of matching the predicted tiger door position was weighted by a parameter beta. Here, it was also assumed that if the observed tiger door position disagreed with the prediction, the participant would re-estimate the grid using the current tiger door prediction (re-estimation), but could continue to update the grid position inferred on the previous trial (which did not match the observation) with probability epsilon. The estimated beta value, which indicated the accuracy of grid extraction from the observations,

was high ($0.97 \pm 0.045$), while the epsilon value, which indicates the probability of dragging incorrect estimate was low ($0.14 \pm 0.29$); these results suggest that the participants tended to infer the hidden state accurately from the observations. On the other hand, if they moved to an adjacent grid, the next grid location was inferred from the structure of the maze, which in turn predicted the tiger door position (Fig. 3a; for further details, please refer to Supplementary Fig. 2 and the "Methods" section, specifically the Behavioural model). In both the Listen and Move trials, the grid location had to be estimated from the memory of the maze structure, but to account for participants' imperfect memory, an error rate gamma was introduced: the mean of estimated gamma was very small ($0.074 \pm 0.057$) and the individual estimates were negatively correlated with the accuracy of the grid location prediction ($r = -0.55$, $p = 1.3 \times 10^{-2}$). We also examined two alternative models: a top-down model that inferred only the grid location (the tiger door position is uniquely derived from the maze structure based on the grid

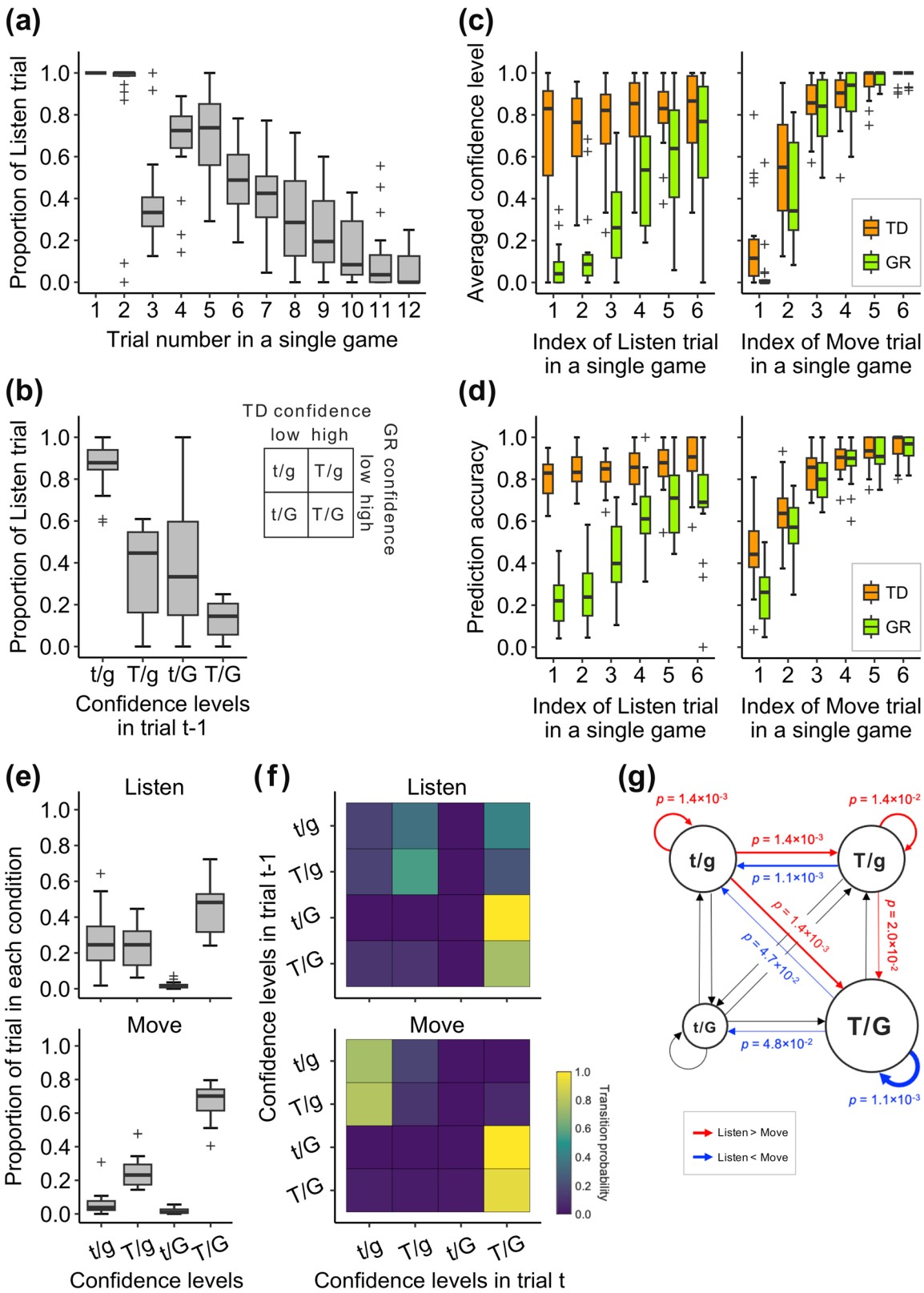

location inference) and a parallel inference model in which the tiger door and grid location are inferred independently (see the "Methods" section, specifically the Alternative models and Supplementary Fig. 2). Random-effects BIC analysis demonstrated that the hierarchical inference model was the best model to explain the participants' behaviour, even when considering the additional complexity of the model (Table 1 and Supplementary

Fig. 3). Our hierarchical inference model accurately reproduced the participants' decisions for both predicting the tiger door position ($87.2 \pm 2.3\%$) and predicting the grid location ($60.9 \pm 6.2\%$). The reproducibility was significantly better than chance, even in trials in which the participants' predictions were incorrect (Wilcoxon signed-rank test; tiger door, $56.5 \pm 6.9\%$, $p = 8.9 \times 10^{-5}$; grid, $33.2 \pm 5.4\%$, $p = 8.9 \times 10^{-5}$).

**Fig. 2 | Behavioural results. a** The proportion of the trials in which the participants chose to listen during the action phase as a function of the trial number per game. Each box extends the lower to upper quartiles, with the median of all subjects as the horizontal line, and the whiskers indicating 1.5× IQR. The cross markers indicate outliers. **b** The participant-wise proportion of the Listen trials, categorised by four (2 × 2 multiplicative) conditions of the confidence levels for the tiger door and grid predictions from the previous trial. Lowercase t and g (or uppercase T and G) represent confidence levels for the tiger door and the grid prediction respectively: lowercase letters signify low confidence, while uppercase letters denote high confidence. Each box extends from the lower to upper quartiles, with the median indicated as the horizontal line, and the whiskers representing 1.5 × IQR. The cross markers denote the outliers. **c** The averaged confidence level for the tiger door and the grid prediction as a function of the number of the Listen (left panel) or Move trials (right panel) in a single game. The confidence in predicting tiger door position and grid location were strongly correlated in the Move trials ($r = 0.94$, $p = 3.5 \times 10^{-58}$). Conversely, the correlations between the confidence levels for these predictions were considerably lower in the Listen trials ($r = 0.47$, $p = 1.1 \times 10^{-7}$). Each box represents the lower to upper quartiles, with the median as the horizontal line, and the whiskers indicate 1.5 × IQR. The cross markers represent outliers. Each data point represents the average accuracy for a single participant. Although the number of the Listen and Move trials varied between games (Listen, maximum 10, mean 5.0 ± 1.6; Move, 9, 4.8 ± 2.1), the Figures show results up to the sixth trial as data with a sufficient number of samples (more than 5 trials) to justify the test (one-sided Wilcoxon signed-rank test, $p < 0.05$). **d** The accuracy for the tiger door and the grid prediction as a function of the number of the Listen (left panel) or Move trials (right panel) in a single game. The prediction accuracy for the tiger door position and grid location were strongly correlated in the Move trials ($r = 0.90$, $p = 1.3 \times 10^{-44}$), and

were considerably lower in the Listen trials ($r = 0.33$, $p = 3.1 \times 10^{-4}$). Each box represents the lower to upper quartiles, with the median as the horizontal line, and the whiskers indicate 1.5×IQR. The cross markers represent outliers. Each data point represents the average accuracy for a single participant. **e** The proportion of the trials categorised by four (2 × 2 multiplicative) conditions based on the confidence levels for the tiger door and grid predictions from the previous trial, in the same manner as (**b**), depending on the selected action in the current trials (top, Listen; bottom, Move). Each box represents the lower to upper quartiles, with the median as the horizontal line, and the whiskers indicate 1.5 × IQR. The outliers are denoted by cross markers. **f** State transition matrices illustrating confidence levels in the tiger door and the grid predictions for both Listen (top panel) and Move (bottom panel) actions. Each square in the heat maps represents the median value, for all participants, indicating the occurrence probability of the confidence condition in trial $t$ (horizontal axis), followed by each action in the confidence condition of trial $t-1$ (horizontal axis). **g** Frequency matrices and a diagram of the state transition calculated from the likelihood of each confidence condition in the previous trial (**e**) and the state transition matrices of the confidence levels (**f**). Each square in the heat maps represents the median frequency across the participants. The diameter of each node in the diagram corresponds to the occurrence probability of the confidence condition (t/g, 25.6 ± 13.1%; t/G, 1.8 ± 1.6%; T/g, 25.2 ± 10.3%; T/G, 47.5 ± 12.3%). The width of the arrows in the diagram corresponds to the median frequency of transitions across the participants, with red arrows indicating that significantly more transitions were observed in the Listen trials than in the Move trials, and blue arrows indicating the opposite (one-sided Wilcoxon signed-rank test, false discovery rate (FDR) corrected). The black arrows indicate that there is no significant difference in the transition frequency between the two actions or that the median transition frequency is zero.

The entropies of the tiger door position and grid location inferences estimated from the hierarchical inference model corresponded well with the confidence levels reported by the participants for their predictions; that is, the entropy was considerably higher when the confidence levels were high (Fig. 3b, Wilcoxon signed-rank test; tiger door, $p = 8.9 \times 10^{-5}$; grid, $p = 8.9 \times 10^{-5}$). In addition, the hierarchical inference model-based entropies were substantially higher before the participants chose to listen in the action phase than before they chose to move (Fig. 3c, Wilcoxon signed-rank test; tiger door, $p = 8.9 \times 10^{-5}$; grid, $p = 8.9 \times 10^{-5}$). The entropies of the tiger door and grid inference displayed similar behaviour to the confidence level when plotted over trials. Specifically, in the Listen trials, only the entropy of the tiger door decreased from the beginning, whereas in the Move trials, both entropies decreased gradually as the number of trials increased (Fig. 3d). The correlation between these two types of entropies was much lower in the Listen trials than in the Move trials (Fig. 3e, $p = 8.9 \times 10^{-5}$). Thus, the model-based entropy derived from the probability distributions estimated from the participants' behaviours alone exhibited properties similar to the metacognitive judgements of the participants' subjective confidence. This suggests that the hierarchical inference model can successfully replicate information processing in the participants' brains.

**Table 1 | Bayesian model comparison results**

| Model | BIC | MF | XP | δ | β | ε | γ |
|---|---|---|---|---|---|---|---|
| Hierarchical inference | 316.8 ± 76.7 | 0.67 | 0.998 | 1.80 ± 0.74 | 0.97 ± 0.014 | 0.14 ± 0.29 | 0.074 ± 0.045 |
| Hierarchical inference without re-estimation | 341.6 ± 80.3 | 0.16 | $1.5 \times 10^{-3}$ | 1.78 ± 0.70 | 0.96 ± 0.043 | – | 0.087 ± 0.071 |
| Parallel inference | 399.5 ± 57.1 | 0.042 | $8.0 \times 10^{-6}$ | 1.86 ± 0.88 | – | – | 0.054 ± 0.066 |
| Top-down inference | 343.4 ± 79.7 | 0.13 | $6.0 \times 10^{-4}$ | – | – | – | 0.031 ± 0.030 |

For each model, the table summarises the fitting performance (*BIC* Bayesian information criteria, *MF* model expected probability, *XP* model exceedance probability) and the estimated model parameters (mean ± SD).

## Brain activity involved in processing the action feedback

The behavioural analysis results suggested that different information processing steps occurred after executing listening to and moving actions during the action phase. This is because the explicit feedback for inference, i.e., the tiger roar, was given only after choosing to listen. In a subtraction analysis comparing brain activity during the action feedback period of the Listen and Move trials, it was observed that the brain regions displaying significantly higher activation during the Listen trials were widely distributed across the cerebrum, including the right rostrolateral prefrontal cortex (rlPFC) and the parahippocampal gyrus (PHG) (Fig. 4a, Supplementary Fig. 4, and Supplementary Table 1). In contrast, no cluster exhibited significantly higher activity during the Move trials than in the Listen trials.

According to our hierarchical inference model, there are two types of information processing in the brain during the Listen trials: one is an updating mode, which makes previous prediction more confident if the position of the tiger roar matches the tiger door position predicted based on the history. The other mode is a re-estimation mode, which re-estimates the grid location prediction if the tiger roar was heard from a different position than predicted. Upon comparing brain activity during the action feedback period, we found that neural activity in the dorsomedial prefrontal cortex (dmPFC) and bilateral anterior insula increased in the re-estimation trials compared to the updating trials (Fig. 4b). Activities involved in updating were observed in the supplementary motor area (SMA) and left fusiform gyrus (Fig. 4c, Supplementary Table 2).

## Neural correlates involved in the uncertainties during the hierarchical inference

To identify the neural activities involved in the two types of uncertainty of the inference, we first performed a general linear analysis in which the two prediction delays in each trial were modulated by the corresponding prediction confidence levels, which was the participants' introspective evaluation of the uncertainties: 0.5 (high) or −0.5 (low). It was found that some cortical areas were negatively correlated with confidence levels (Fig. 5a, b, cool colour scale, Supplementary Table 3). We also constructed a general linear model (GLM) using the two types of posterior entropies estimated using our hierarchical inference model. The brain regions that exhibited

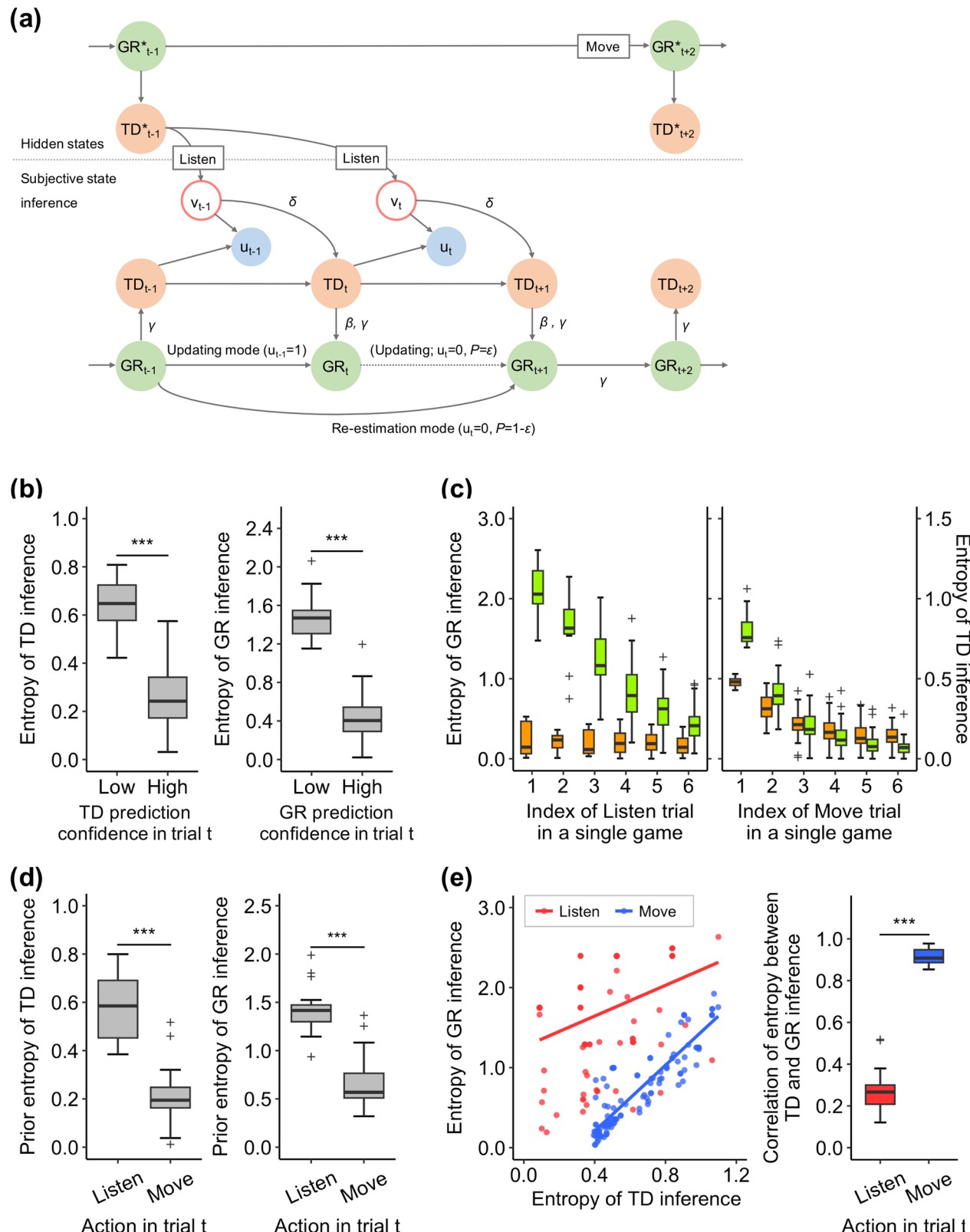

activity correlated with these posterior entropies included not only the regions of interest associated with the corresponding subjective confidence levels, but also extended to other areas (Szymkiewicz-Simpson coefficient for the tiger door confidence and entropy, SS = 0.80; for the grid confidence and entropy, SS = 0.85; see also Fig. 5a, b, warm colour scale, and Supplementary Table 4). For the tiger door inference, the bilateral insula, thalamus,

and putamen exhibited activity that was correlated with increasing entropy (Fig. 5a). In contrast, BOLD activity in the bilateral thalamus, PHG, and putamen was positively correlated with grid inference entropy (Fig. 5b). Both the dmPFC and putamen displayed increased activity that correlated with both tiger door position entropy and grid entropy; however, the localisation was different, that is, the activity associated with tiger door

**Fig. 3 | Hierarchical Inference model and model-based behavioural analysis results. a** Graphical description of the hierarchical inference model. This model infers the probability distribution of the upcoming tiger door position and grid location based on the sequence of the direction of the participants' movement and the position of the roaring door ($v_t$). This model assumes that when the participants chose to listen, they first infer the upcoming tiger door position, and subsequently estimate the upcoming grid location based on that information using a different prior distribution depending on whether the predicted tiger door matched the direction of the observed tiger roar (updating mode, $u_t = 1$) or not (re-estimation mode, $u_t = 0$). Conversely, when they choose to move, they first infer the grid location and then derive the tiger door position. The four variables described alongside the edges (delta, beta, epsilon and gamma) are the free parameters that modulate the inference process of the hierarchical inference model. For further details, please refer to Supplementary Fig. 2a and the "Methods" section, specifically the Behavioural model. **b** Mean posterior entropy of the tiger door position inference (left panel) and the grid location inference (right panel) compared with participants' reported confidence levels. In both cases, the entropy was significantly higher for the trials with low confidence compared to trials with high confidence (Wilcoxon signed-rank test, ***$p < 0.001$). Each box extends from the lower to upper quartiles with a horizontal line at the median. The whiskers indicate $1.5 \times$ IQR, and cross markers represent the outliers. **c** Mean posterior entropy of the tiger door inference (orange boxplot) and the grid inference (green boxplot) as a function of the number of the Listen trials (left panel) and/or the number of the Move trials (right panel) in a single game. In the Move trials, both entropies decreased as the trials progressed and were correlated with each other. Conversely, in the Listen trials, the entropy of the grid inference decreased, while the entropy of the tiger door inference remained low from the beginning, and these entropies were not correlated. **d** Mean prior entropy of the tiger door inference (left panel) and the grid inference (right panel), depending on the action chosen by the participant in the action phase. For both inferences, the prior entropy was significantly higher when participants chose to listen than when they chose to move (Wilcoxon signed-rank test, ***$p < 0.001$). Each box represents the lower to upper quartiles, with a horizontal line at the median. The whiskers indicate $1.5 \times$ IQR, and cross markers represent the outliers. **e** An example plot illustrating the correlation between the posterior entropies of the tiger door and the grid inferences in the Listen and Move trials for an example participant (left panel) and its mean value across all participants (right panel). The correlation between the two types of uncertainty was lower in the Listen trials compared to the Move trials (Wilcoxon signed-rank test, ***$p < 0.001$). Each box extends from the lower to upper quartiles, with a horizontal line at the median. The whiskers indicate $1.5 \times$ IQR, and the cross markers represent the outliers.

entropy was situated more posteriorly in both regions, while the activity associated with grid entropy was primarily observed in the anterior part (Fig. 5c).

## Discussion

In this study, we designed a Tiger maze navigation task to test how the brain enables us to estimate our own location and that of a tiger within a maze based on probabilistic information. We proposed a novel hierarchical model as the generative model of hidden-state navigation, in which our grid location in the maze and the tiger door position interact hierarchically. Our model fits the participant's prediction performance better than alternative parallel or top-down inference models. The results of the behavioural data analysis revealed that the uncertainty associated with inferring higher level hierarchical hidden state, grid location, and lower-level state, tiger door position, was resolved over different time scales, and these processes were effectively replicated by our hierarchical inference model (Fig. 3, Supplementary Fig. 3).

In our proposed model, the order of inference depends on the type of action performed before it; i.e., after the listening action (where participants can gather probabilistic information about the tiger door position), the tiger door position is inferred first, with the grid location later inferred on that basis. Conversely, after the moving action (where participants transition between grid locations), the inference is performed in the opposite direction. Note that most previous navigation researches have formulated the human behaviours with the framework of reinforcement learning and goal-directed planning[27,28,30]; in our study, however, the navigation behaviours were not modelled primarily due to the unknown reward function to participants and lacking a particular goal state. In most hierarchical models proposed to date, low-level inference has been unidirectionally controlled by high-level inference[10,12,17,32]; however, such a model failed to adequately explain participants' behaviour in the task (Table 1, Supplementary Fig. 3). The interactive information flows between the hierarchical hidden states, as introduced in our model, have been proposed in machine learning studies, particularly in the context of hierarchical neural networks featuring bidirectional information flows[28–30], suggesting its potential significance for future endeavours in replicating and understanding complex human information processing. Our hierarchical inference model has the potential to be applied to cognitive problems involving hierarchies, such as transfer learning and social interactive decision-making.

Most previous studies have dealt with maze environments aimed at reaching a goal state, and have formulated navigation behaviour within the framework of reinforcement learning and goal-directed planning[27–29]. However, our Tiger maze navigation task is a problem setting that focuses on hidden-state inference and the goal state is unknown, therefore the participants' navigation behaviour was used as an observed variable rather than being modelled directly by a generative model. Although not introduced in our hierarchical inference model to avoid complication, it would be possible to model the navigation behaviour based on the goal of exploring as many unvisited states as possible to avoid opening the tiger door. Specifically, behaviour in our task could be modelled as a hierarchy of two stages: the decision to listen or move and, if moving, the direction in which to move. The former could be formulated as an approach-avoidance conflict model[33,34], where the conflict is between the avoidance behaviour of avoiding the tiger door and the exploration behaviour of reaching the goal. The choice of moving direction could then be determined by the objective function of maximising the explored grids in the maze[20].

During the action phase, several brain regions across the cerebral cortex, including the rlPFC and PHG, exhibited significantly greater activation during the Listen trials than in the Move trials (Fig. 4a). Conversely, no regions displayed significantly higher activity in the opposite contrast (Supplementary Fig. 4, Supplementary Table 1). This finding is reasonable as the cognitive effort is higher during the Listen trials because they provide explicit feedback for updating the inferences. Moreover, the feedback itself, presented as a visual stimulus is also more complex, with the listening action more likely to be selected when the confidence level is low. The rlPFC is thought to be activated in contexts involving exploration based on uncertain or probabilistic information[35,36] and updating the possibilities of multiple candidate grids simultaneously[37–39], from which the most likely option is selected[40,41]. Updating grid location inference requires spatial mental simulation within the memorised maze, a process likely to involve PHG activity, known for processing spatial location information[42–44].

The computational analysis using the model parameters allowed for a more refined assessment of the possible roles of different brain regions. Consequently, the imaging analysis employing the hierarchical model can identify brain areas related to specific inference processes during the action phase. In the trials where the hierarchical inference model estimated that the grid location needed re-estimation due to the feedback (that is, tiger roar position) differing from the prediction, activity in the dmPFC and bilateral anterior insula increased (Fig. 4b, Supplementary Table 2). This re-estimation process involves changing one's beliefs about their position in the maze. Notably, activity in the dmPFC has been previously observed with the re-estimation of hidden states in a comparable partially observable maze navigation task[19]. This area is known to play a role in switching behaviour[45–47] due to conflict[48], error detection[49,50] and error prediction[51,52]. Hence, we predicted this result a priori. We also note that the anterior insula is known to encode prediction errors in decision-making[53–56], although this was not one of our a priori predictions.

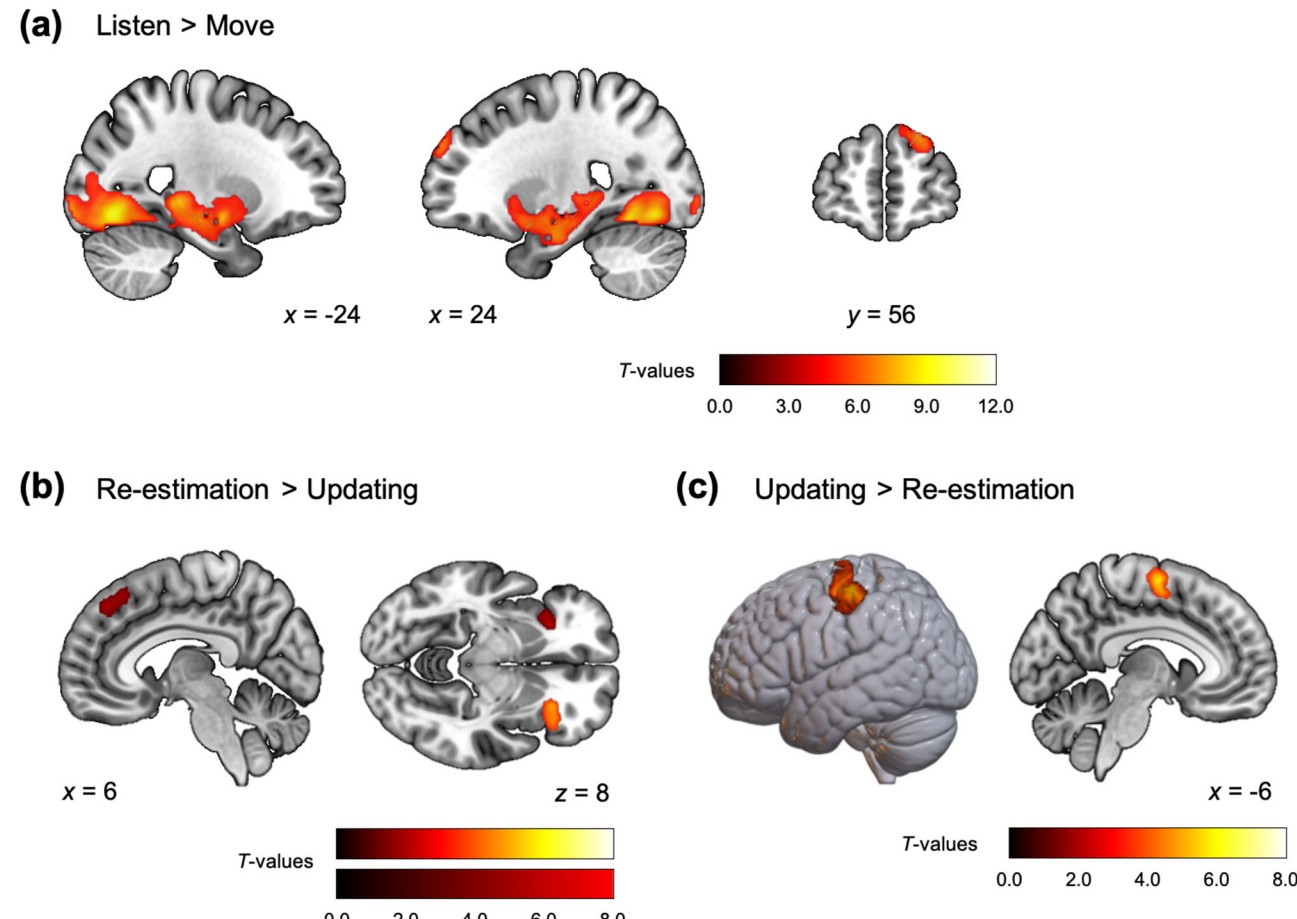

**Fig. 4 | Brain activation involved in processing the action feedback. a** Brain areas that exhibited greater activity at the onset of the action feedback in the Listen trials than in the Move trials. The clusters are significant at $p < 0.05$ family-wise error (FWE)-corrected, with the cluster-defining threshold $p < 0.0001$ uncorrected (refer also to Supplementary Fig. 4). **b** Brain activation regions involved in processing during the re-estimation mode at the onset of the action feedback in the Listen trials

(step 3a in Fig. 1a). The clusters are significant at $p < 0.05$ FWE-corrected, with the cluster-defining threshold $p < 0.001$ uncorrected (Note that the clusters indicated with the red colour bar correspond to $p < 0.05$ uncorrected in the cluster level). **c** Brain activation regions involved in processing during the updating mode at the onset of the action feedback (step 3a, b in Fig. 1a). The clusters are significant at $p < 0.05$ FWE-corrected, with the cluster-defining threshold $p < 0.001$ uncorrected.

On the other hand, when the feedback was as predicted, the previous inference (that is, belief about the grid location) was judged to be correct and was updated. Imaging analysis revealed that this was associated with greater activity in the SMA in these trials. In these trials, the participants indeed tended to select the same options as their estimated tiger door position in previous trials ($76.5 \pm 6.9\%$ after the updating mode; $31.0 \pm 10.3\%$ after the re-estimation mode; $p = 8.9 \times 10^{-5}$). This illustrates a clear functional distinction between dmPFC and SMA responses in the context of this task.

We also compared brain activity in relation to the uncertainty of inference in the prediction phase, both in terms of subjective evaluation (confidence) and entropy estimated by the hierarchical inference model (Fig. 5a, b). The dACC and precentral gyrus activity, whose activities are correlated with both uncertainty indices, have also been reported in previous metacognition studies to be brain regions that provide metacognitive control signals (i.e., confidence) in various types of decision-making[57–59] and learning[11]. Subjective confidence levels were based on the participants' introspective and metacognitive evaluation of the uncertainty of the hidden-state inference reported in a binary form (high or low confidence). In other words, the coarse information extracted from inferences may not be a sufficiently accurate measure for capturing the neural correlates of the inference itself, which can be expressed as a probability distribution. As previous studies suggested that the posterior probability distribution is represented as the multivoxel activity patterns in the localised brain

regions[60,61], in these regions the inferences may be encoded across multiple neuronal populations. Our proposed hierarchical inference model can compute the posterior distribution of state inference from which subjective confidence is derived. Its entropy is a measure of uncertainty regarding inference, which is, therefore, negatively correlated with confidence. Consistent with this model, the brain regions identified by parametric modulation analysis using the two types of entropy (entropy pertaining to tiger door position and grid location; Fig. 5a, b, warm colour scale, Supplementary Table 4) fully encompassed the brain regions that were negatively correlated with confidence levels for both tiger door and grid location predictions (Fig. 5a, b, cool colour scale, Supplementary Table 3).

For both types of inference uncertainty, the subregions of the parietal region exhibited increased activity when confidence was low compared to when confidence was high; however, the active regions differed depending on the type of information being predicted. For tiger door prediction, the participants inferred the hidden state as being in one direction (left, front, or right) based on their position, and we found that the inferior parietal lobule activity displayed orientation sensitivity[62,63] when there were multiple directional possibilities (Fig. 5a). In contrast, the right superior parietal lobule, which is associated with spatial navigation[64–66] demonstrated higher activity when confidence in predicting the grid location in the maze was low (Fig. 5b). The parietal cortex is known to be involved in spatial navigation[67–69] and these data add to the growing understanding of how it supports computationally precise functions.

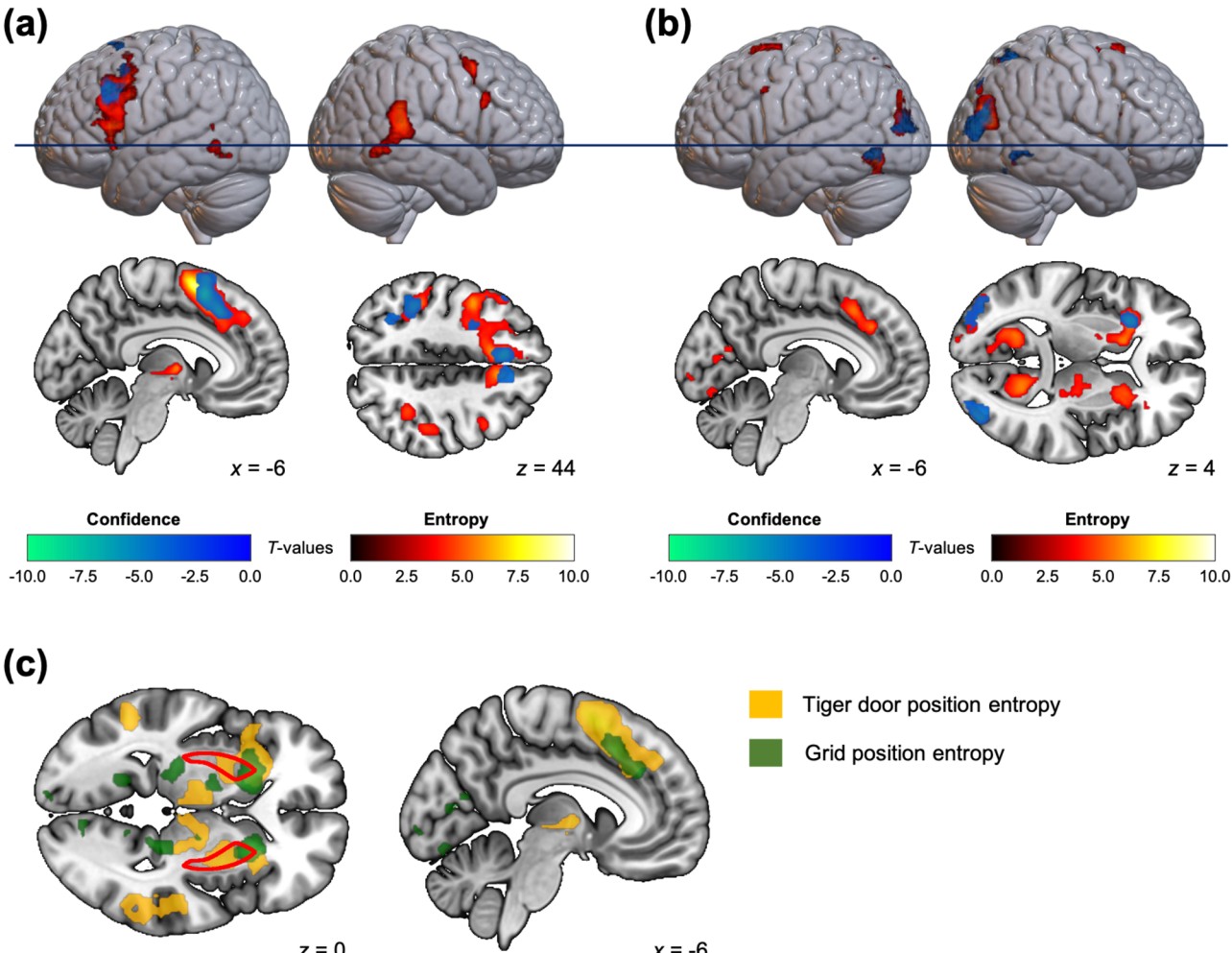

**Fig. 5 | Neural correlates involved in uncertainties in hierarchical inference.**
Neural representation of the posterior entropy for the tiger door position inference (**a**) and those for the grid location inference (**b**) during the corresponding prediction delay (steps 4 and 8 in Fig. 1a). The overlapping contrasts (green to blue) correspond to those of the confidence levels for the tiger door position prediction (**a**) or for the grid location prediction (**b**). Note that the confidence-related clusters shown in the Figures were the brain areas which exhibited negative activity at high confidence, i.e., brain regions that are more active when the prediction is less confident. There were no brain regions that displayed activity negatively correlated with the entropies, and no regions that showed activity positively correlated with confidence level. The clusters are significant at $p < 0.05$ FWE-corrected, with the cluster-defining threshold $p < 0.001$ uncorrected. The blue horizontal line indicates the axial slice at $z = 0$, as shown in **c**. **c** Overlap between brain regions whose activity increased with increasing tiger door position entropy (yellow) and the grid location entropy (green). The bilateral putamen is indicated by the areas encircled by red lines.

We found that the dmPFC and dACC were active when hidden-state inferences (beliefs) tended to be updated. This observation is consistent with previous studies on humans[70–72], non-human primates[73] and rodents[74], and supports the idea that the dACC is active in uncertain environments where the internal models were updated based on new observations[23,71]. The hierarchical nature of the hidden states in this study suggests that different subregions of the dmPFC become active at different levels of the hierarchy, i.e., a higher level hidden state (grid location) is associated with a more anterior part of the dmPFC. Previous studies have indicated that control abstraction is hierarchically organised along the rostral axis of this area, with more caudal parts involved in lower-order behavioural control, and more rostral parts involved in higher-order strategic control[75–78]. This finding is also supported by research using the neurocomputational theory, which involves planning control over multiple levels of abstraction[79–81].

Similarly, in the bilateral putamen, the anterior areas were active for the grid inference, which is a higher level hidden state in the hierarchy, and the posterior areas were active for tiger door inference. To infer the grid location, it is necessary to integrate both uncertain (probabilistic) information and mental simulations based on the maze structure in the memory (internal world model). In contrast, the tiger door position was inferred solely from externally given observations and did not require an internal model. In other words, these inference processes are based on model-based or model-free control, respectively, which is consistent with previous results that suggest the neural circuitry involved in the anterior and posterior putamen are functionally separable, with the anterior corresponding to associative, goal-directed, model-based control, and the posterior to sensorimotor, habitual, model-free control in human[82–85], non-human primates[86,87], and rodents[88,89].

One notable strength lies in the innovative use of computational modelling, which allowed us to reproduce human hierarchical inference processes as the spatial inference problem in the navigation task, and to extract the neural substrates involved in inference using both subjective (i.e., confidence) and objective measures (uncertainty). However, it is important to acknowledge that cognitive research using models has its limitations. In our case, the computational model was explicitly hierarchical since the task presented an explicit hierarchy of states, whereas in a real-world environment, it is natural to assume that humans' hierarchies or subdivide problems themselves based on the structure of states; this may overlook complex and important mechanisms of cognitive phenomena. In addition, conclusive evidence concerning the existence of functional hierarchical information

processing among these neural substrates has not been presented. It has also not been demonstrated how the model parameters estimated for each individual are related to their neural activity. In our model, the parameters were involved in modifying the mechanisms that integrate observation and the two types of inference and may control the functional connectivity between these neural substrates. In this study, however, these hypotheses were not assessed mainly due to the limitations of the temporal resolution of the fMRI signals and the event-related task design. An intriguing avenue for future research lies in exploring the hierarchical functional networks within the brain and their correspondence with the computational models, potentially by employing higher temporal resolution measurements and analytical methods, and by using intervention-based approaches to verify causality in the brain network.

In summary, in this study, we designed a Tiger maze navigation task that combines two typical POMDP tasks: the Tiger problem and the partially observable maze navigation. We proposed a computational model of behaviour that frames behaviour as a spatial inference problem characterised by hierarchical hidden states. Brain imaging analysis has suggested that inference based on information under conditions of multi-layered uncertainty engages different areas of the medial prefrontal cortex and the basal ganglia, depending on the inference's levels in the hierarchy and the nature of inference (model-based or model-free). Additionally, our imaging results demonstrated that the metacognitive evaluations of uncertainty, or confidence, in the context of hierarchical inference are represented in a cortical frontoparietal network. This enriches our understanding of the neural architecture of complex decision-making processes.

## Methods

### Subjects

In this study, twenty healthy participants (6 females; aged 20–29 years) were recruited for the experiment, and written informed consent was obtained. This study was approved by the ethics committees of the Advanced Telecommunications Research Institute International, Japan and the Graduate School of Informatics, Kyoto University, Japan. All ethical regulations relevant to human research participants were followed. Prior to undergoing functional magnetic resonance imaging (fMRI) scanning, all participants underwent a pre-experimental training task and engaged in a behavioural experiment concerning the Tiger maze exploration task. The minimum number of participants was defined as 18, based on a power analysis ($\alpha = 0.05$, $1-\beta = 0.8$), with the effect size calculated from a previous study on hierarchical decision-making[18] using G*Power (http://www.gpower.hhu.de/).

### Tiger maze navigation task

Participants explored a virtual $4 \times 4$ grid maze, where each grid had doors on all four sides, one of which had a tiger behind it (tiger door) that should not be opened (Fig. 1). The maze was designed with topological connections, resembling a torus, connecting the upper (north) and lower (south) boundaries of the maze as well as the left (west) and right (east) boundaries. We used a single common maze for all participants (Fig. 1b) to avoid individual variability in task difficulty. Since both tiger doors and normal doors have an identical appearance, the three-dimensional (3D) scene that participants can observe in the maze (three identical doors on the front, left, and right walls) is the same in all grids. In other words, the maze is partially observable, and the current observation alone does not determine the current grid location.

Each game starts from an unknown initial state, with each trial in the game divided into two phases: an action phase for exploring the maze and a prediction phase for reporting predictions (Fig. 1a). In the action phase, a 3D scene of the current location and body orientation was first displayed, and participants chose to either move (one of three moves: left, forward, or right) or listen (Fig. 1a). If they chose to move through a normal door, a short animation of moving to the selected adjacent grid was displayed. However, if they chose to move to the tiger door, the game ended. When participants chose to listen, a tiger roar was heard from one of the three doors, which turned red and displayed a letter indicating direction (L, F, or R). The accuracy of the tiger roar observation was probabilistic, with an 85% chance of a roar being observed from the tiger door and 7.5% from the normal door, with participants being informed of these probabilities. If no action was selected within 4 s, a fixation point was presented for 1.5 s, and the state remained the same until the next trial ($0.35 \pm 1.1$ trials).

In the prediction phase, participants were first asked to predict the position of the tiger door in the next trial and rate their confidence in the prediction. The tiger door prediction was chosen from the three letters indicating the door position (L, F, and R), and the confidence level was chosen from two options: high (H) and low (L), with the positions of the options set randomly for each trial. Participants were then asked to predict the position of the grid in the maze in the next trial and rate their confidence level, as in the case of the tiger door. The prediction of the grid location was reported by selecting the corresponding row and column coordinates on the maze picture in order: the row coordinate (1: top, to 4: bottom), and then the column coordinate (1: left, to 4: right). All choices were required to be made within 2 s, and no choice trials were excluded from the analysis as missed trials ($6.4 \pm 5.4\%$). Each prediction and confidence report were preceded by a delay (4 s), with a fixation cross to allow participants to prepare their choices mentally. During the delay before grid prediction, a $4 \times 4$ grid was overlaid to encourage mental imagery of the grid location. For both predictions and confidence choices, feedback was provided as a green frame around the chosen option for 1.5 s. However, this feedback did not indicate whether the prediction was correct or not.

The game termination conditions were determined based on participants' performance. These conditions were defined as having visited more than eight grids or having completed a specified number of trials (10–14 trials). If participants met these conditions, a yellow star was displayed on one of the doors, signifying the achievement of their goal, and the game score was displayed. If participants chose a tiger door, the game concluded with a score as '0'. Each game comprised 2–14 trials, with an average of $10.9 \pm 2.4$ trials per game. Participants were instructed that the more grids they explored and the more correct their predictions were, the more points they would receive (see "Game score" section). Each participant completed a total of 24 games ($237.2 \pm 22.0$ trials) divided into eight sessions: four sessions (12 games) in the behavioural experiment outside the scanner and another four sessions (12 games) in the fMRI scanning experiment, with both experiments performed on the same day.

### Game score

The game score was defined by participant's performance in both the action and prediction phases as follows:

$$score = \left\{ N_{exp} \times 5 - N_{lis} + \sum_{t=1}^{T-1} \left( rw_{TD,t} + rw_{GR,t} \right) \right\} \quad (1)$$

For the action phase, rewards were added according to the number of grids visited, $N_{exp}$, with a small cost added according to the number of times Listen was chosen, $N_{lis}$, during the game. The score for the prediction trials, both for the tiger door prediction $rw_{TD,t}$, and the grid prediction $rw_{GR,t}$ was set to be higher when the correct prediction was made with high confidence (see Supplementary Table 5), and was summed over all prediction trials $T$ in the game. The average scores for the 12 games were $37.8 \pm 9.5$ and $37.9 \pm 7.6$ in the behavioural and scanning experiments, respectively. The participants were paid a base monetary reward and an incentive based on the number of points they had scored ($4700 \pm 1100$ yen).

### Training task

Approximately one week before the main experiment, participants performed a training task outside the scanner, in which they learned the task procedure and the structure of the maze. First, participants performed the Tiger maze navigation task, as in the main experiment, for 1 h and 45 min, but with reference to a printed two-dimensional maze map. Subsequently,

they performed the same task without a printed map. Participants were compensated with a base payment and no performance-based rewards in the training task. Note that we did not model the participants' decision-making behaviours, but used their actions as an observed variable for modelling the hidden-state inference.

## Behavioural model

We proposed a computational model of participant behaviour using Bayesian methods, namely, a hierarchical inference model. In this model, the order in which the two hidden states are inferred varies depending on the chosen actions (Listen or Move); that is, inferences are made through a bidirectional flow of information from the higher or lower levels of the hierarchy. This model simulates the generative process of the probabilities associated with each option for predicting tiger door position $s_{TD}$ ($|s_{TD}| = 3$) and grid location $s_{GR}$ ($|s_{GR}| = 16$) based on the sequence of observable variables: participants' actions ($a$) and the position of the roaring door ($v$). Throughout the model, variables marked with an asterisk (*) indicate true values, while variables marked with a hat (^) are subjective and inferred by the model. Here, the sequences of the participants' actions were regarded as the observation variables for this model.

If the Listen action was selected in trial $t$, the position of the tiger door in trial $t + 1$ was first determined as follows (step 3a in Supplementary Fig. 2a):

$$P_{t+1}\left(s_{TD,t+1}\right) \propto P_t\left(s_{TD,t}\right) L\left(s_{TD,t}|v_t\right)^\delta \quad \text{where} \quad L\left(s_{TD,t}|v_t\right) = \begin{cases} \alpha & \text{if } s_{TD,t} = v_t \\ \frac{1-\alpha}{2} & \text{otherwise} \end{cases} \quad (2)$$

where $\alpha$ is the probability of roar observation from a tiger door (0.85) and $\delta$ is the sensitivity parameter that exponentially increases the influence of the information from the new observation. When the observed position of the roaring door ($v_t$) agreed with the tiger door prediction ($\hat{s}_{TD}$), participants credited the history of the observations in the current grid and updated the grid location probabilities. If participants perfectly memorised the maze structure, the grid location probabilities would be updated in the form of Bayesian filtering as follows:

$$P_{t+1}\left(s_{GR,t+1}|v_{t':t}, \hat{s}_{TD,t+1}\right) = P_{UD,t+1}\left(s_{GR,t+1}|v_{t':t}, \hat{s}_{TD,t+1}\right)$$
$$\propto \begin{cases} \beta P_t\left(s_{GR,t}|v_{t':t}\right) & \text{if } s^*_{TD}\left(s_{GR,t}, d_t\right) = v_t \\ (1-\beta) P_t\left(s_{GR,t}|v_{t':t}\right) & \text{if } s^*_{TD}\left(s_{GR,t}, d_t\right) \neq v_t \\ 0 & \text{if } s^*_{TD}\left(s_{GR,t}, d_t\right) = \text{back} \end{cases} \quad (3)$$
$$\text{where} \quad \hat{s}_{TD,t+1} = \text{argmax}_{s_{TD}} P_{t+1}\left(s_{TD,t+1}\right)$$

$s^*_{TD}$ is the true tiger door position in the grid $s_{GR}$ with an observable body orientation $d_t$. $\beta$ is the degree of dependence on the tiger door prediction; if $\beta = 1$, participants extract grids for which $s^*_{TD}\left(s_{GR}, d_t\right)$ is consistent with $s_{TD}$ as the candidates and consider all others as unlikely. $t'$ denotes the number of the first trials after transfer to the current grid $s^*_{GR,t}$. Because the door behind participants was always passable, grids where $s^*_{TD}\left(s_{GR}, d_t\right)$ was the backside should be excluded from the candidates of the grid location (the third case of Eq. (3)). Here, if the participants' memory incomplete, i.e., they erred in recalling the maze structure with the probability $\gamma$, they mistakenly update the grid location probability according to the second case in Eq. (3), even when the true tiger door position matched the roaring door position (corresponding to the first case), with the error probability $\gamma$: similarly when the true tiger door position was inconsistent to the roaring door position (corresponding to the second case), the update in the first case in Eq. (3) could occur with $\gamma$. Participants also could update the grid location where $s^*_{TD}\left(s_{GR}, d_t\right)$ was backside if they made a mistake about the maze structure (corresponding to the third case). In summary, the dynamics of the grid inference are defined as follows (update mode, step 4a

+ in Supplementary Fig. 2a):

$$P_{t+1}\left(s_{GR,t+1}|v_{t':t}, \hat{s}_{TD,t+1}\right) = P_{UD,t+1}\left(s_{GR,t+1}|v_{t':t}, \hat{s}_{TD,t+1}\right)$$
$$\propto \begin{cases} \left[(1-\gamma)\beta + \gamma(1-\beta)\right] P_t\left(s_{GR,t}|v_{t':t}\right) & \text{if } s^*_{TD}\left(s_{GR,t}, d_t\right) = v_t \\ \left[\gamma\beta + (1-\gamma)(1-\beta)\right] P_t\left(s_{GR,t}|v_{t':t}\right) & \text{if } s^*_{TD}\left(s_{GR,t}, d_t\right) \neq v_t \\ \gamma P_t\left(s_{GR,t}|v_{t':t}\right) & \text{if } s^*_{TD}\left(s_{GR,t}, d_t\right) = \text{back} \end{cases}$$
$$\text{where} \quad \hat{s}_{TD,t+1} = \text{argmax}_{s_{TD}} P_{t+1}\left(s_{TD,t+1}\right)$$
$$(4)$$

In contrast, if $v_t$ was inconsistent with $\hat{s}_{TD,t}$, participants rejected the inference $P_t\left(s_{GR,t}\right)$ based on the previous observations $v_{t':t}$, and re-estimated under the current observation $v_t$ and the probabilities at trial $t'$ as the prior distribution (re-estimate mode, step 4a in Supplementary Fig. 2a). Here, it is assumed that even in the re-estimation, participants may (with a certain probability $\varepsilon$) continue to update their inferences based on the previous (unreliable) observations. In summary, the dynamics of the grid inference when $\hat{s}_{TD,t} \neq v_t$ are defined as follows:

$$P_{RE,t+1}\left(s_{GR,t+1}|v_{t':t}, \hat{s}_{TD,t}\right)$$
$$= \begin{cases} \left[(1-\gamma)\beta + \gamma(1-\beta)\right] P_{t'}\left(s_{GR,t'}\right) & \text{if } s^*_{TD}\left(s_{GR,t}, d_t\right) = \hat{s}_{TD,t+1} \\ \left[\gamma\beta + (1-\gamma)(1-\beta)\right] P_{t'}\left(s_{GR,t'}\right) & \text{if } s^*_{TD}\left(s_{GR,t}, d_t\right) \neq \hat{s}_{TD,t+1} \\ \gamma P_{t'}\left(s_{GR,t'}\right) & \text{if } s^*_{TD}\left(s_{GR,t}, d_t\right) = \text{back} \end{cases}$$
$$(5)$$

$$P_{t+1}\left(s_{GR,t+1}|v_{t':t}, \hat{s}_{TD,t+1}\right)$$
$$= (1-\varepsilon) P_{RE,t+1}\left(s_{GR,t+1}|v_{t':t}, \hat{s}_{TD,t+1}\right) + \varepsilon P_{UD,t+1}\left(s_{GR,t+1}|v_{t':t}, \hat{s}_{TD,t+1}\right) \quad (6)$$

Note that, Eq. (5) derived from Eq. (4), in which the prior probability term $P_t\left(s_{GR,t}|v_{t':t}\right)$ was replaced with $P_{t'}\left(s_{GR,t'}\right)$. The grid location was inferred using a method that depended on whether the direction of the observed tiger roar matched their prediction or not. Here, the MAP estimate (a single most likely direction) was used for the tiger door position, rather than the fully probabilistic estimate. This is because the TD observation information (the direction of the door in which the tiger roars) is inherently probabilistic, and there is only one true tiger door state.

If the participants chose to move to an adjacent grid, the inferences were made in the opposite direction, i.e., the probabilities of the grid location were updated based on the memory of the maze structure, and then the tiger door position was predicted based on that inference (steps 3b and 4b in Supplementary Fig. 2a):

$$P_{t+1}\left(s_{GR,t+1}\right) = P_t\left(s_{GR,t}\right) \quad \text{where} \quad s_{GR,t+1} = T\left(s_{GR,t}, d_t, a_t\right) \quad (7)$$

$$P_{t+1}\left(s_{TD,t+1}|d_{t+1}\right) = \sum_{s_{GR,t+1}} P\left(s_{TD}|s_{GR,t+1}, d_{t+1}\right) P_{t+1}\left(s_{GR,t+1}\right)$$
$$\text{where} \quad P\left(s_{TD}|s_{GR,t+1}, d_{t+1}\right) = \begin{cases} 1-\gamma & \text{if } s_{TD} = s^*_{TD}\left(s_{GR,t+1}, d_{t+1}\right) \\ \frac{\gamma}{3} & \text{otherwise} \end{cases}$$
$$(8)$$

$T$ is a fixed transition function, where $s' = T(s, d, a)$ indicates that if door $a$ is selected in grid $s$, facing direction $d$, it will move to grid $s'$.

## Alternative models

We examined two additional models that assume different inference processes. The first is a top-down inference model, which first infers the grid location on trial $t + 1$ and then predicts the tiger door position based on the inference of the grid location and the memory of the maze structure (Eq. (8)), regardless of the chosen action. In the Listen trials, the grid location was estimated from the observed tiger roar direction $v_t$ using incremental

Bayesian filtering, as follows:

$$P_{t+1}(s_{GR,t+1}|v_t) \propto \begin{cases} [(1-\gamma)\alpha + \gamma(1-\alpha)]P_t(s_{GR,t}) & \text{if } s_{TD}^*(s_{GR,t},d_t) = v_t \\ [\gamma\alpha + (1-\gamma)(1-\alpha)]P_t(s_{GR,t}) & \text{if } s_{TD}^*(s_{GR,t},d_t) \neq v_t \\ \gamma P_t(s_{GR,t}) & \text{if } s_{TD}^*(s_{GR,t},d_t) = \text{back} \end{cases} \tag{9}$$

In the Move trials, the grid and tiger door positions were inferred using the same method as in the hierarchical model (Eqs. (7) and (8), respectively).

The second is a parallel inference model which infers the grid location and the tiger door position independently. In this model, the grid location was inferred in the same manner as in the top-down inference model (Eq. (9) in the Listen trials and Eq. (7) in the Move trials). The tiger door position was inferred according to Eq. (2) in the Listen trials. In the Move trials, it is estimated independently of the grid location probability distribution, i.e., based on the probability distribution of the tiger door and the direction chosen in trial $t$ and the maze structure information, as follows:

$$P_{T+1}(s_{TD,t+1}|d_{t+1},a_t) \propto \sum_{s_{TD,t}=\{L,F,R\}} P'(s_{TD,t})P(s_{TD,t+1}|s_{TD,t})$$
$$= \sum_{s_{TD,t}=\{L,F,R\}} P'(s_{TD,t}) \frac{N(s_{TD,t} \to s_{TD,t+1})}{\sum_{s_{TD,t+1}=\{L,F,R\}} N(s_{TD,t} \to s_{TD,t+1})} \tag{10}$$
$$\text{where} \quad P'(s_{TD,t}) = \begin{cases} P_t(s_{TD,t}) & \text{if } s_{TD,t} \neq a_t \\ 0 & \text{otherwise} \end{cases}$$

$$N(s_{TD,t} \to s_{TD,t+1}) = n(S)$$
$$\text{where} \quad S = \{s_{GR}|s_{TD}^*(s_{GR},d_t) = s_{TD,t} \cap s_{TD}^*(s'_{GR},d_t,a_t) = s_{TD,t+1}\} \tag{11}$$

## Parameter estimation and model validation

Our proposed hierarchical inference model has four parameters: the sensitivity to the new evidence in the tiger door position inference ($\delta$), the tiger door prediction dependency of the grid inference ($\beta$), the updating probability in the re-estimation mode ($\varepsilon$), and the imperfectness of subjects' memory of the maze structure ($\gamma$). The parameter ranges were predetermined as $\delta = [1,3]$, $\beta = [0.5, 0.999]$, $\varepsilon = [0,1]$, and $\gamma = [0, 0.3]$. The model parameters were estimated individually by minimising the negative log evidence (Eqs. (12)–(14)):

$$NLE_{TD} = -\log \prod_{g=1}^{G} P(\hat{S}_{TD,g}^*|\boldsymbol{\theta}) = -\log \prod_{g=1}^{G} \prod_{t=1}^{T(g)} P(\hat{s}_{TD,t}^*|\boldsymbol{\theta}) \tag{12}$$

$$NLE_{GR} = -\log \prod_{g=1}^{G} P(\hat{S}_{GR,g}^*|\boldsymbol{\theta}) = -\log \prod_{g=1}^{G} \prod_{t=1}^{T(g)} P(\hat{s}_{GR,t}^*|\boldsymbol{\theta}) \tag{13}$$

$$NLE_{tot} = \eta_{TD}NLE_{TD} + \eta_{GR}NLE_{GR} \quad \text{where } \eta_{TD} = \frac{1}{\log(|s_{TD}|)}, \eta_{GR} = \frac{1}{\log(|s_{GR}|)} \tag{14}$$

Here, $G$ is the number of games, and $\hat{S}_{TD,g}^*$ and $\hat{S}_{GR,g}^*$ are the sequences of the tiger door and grid prediction reported by the participants, respectively, where $T(g)$ is the number of trials in the game $g$. The set of the model parameters is denoted as $\boldsymbol{\theta}$. $\eta_{TD}$ and $\eta_{GR}$ are scaling parameters that compensate for differences in the size of the state space between $s_{TD}$ and $s_{GR}$; $|s_{TD}|$ (=3) and $|s_{GR}|$ (=16) are the number of possible options for the tiger door and the grid, respectively. BIC was used for Bayesian model selection (Table 1). To avoid data circularity, we used data from 12 games in the behavioural experiment for parameter selection and data from the scanning experiment for model validation. When comparing the proposed hierarchical inference model with the alternative models (the top-down inference model and the parallel inference model), we used the parameter values

estimated for each model using Eqs. (12)–(14) ($\boldsymbol{\theta} = \{\gamma\}$ for the top-down model; $\boldsymbol{\theta} = \{\delta, \gamma\}$ for the parallel model).

In validating our proposed hierarchical model, we also used the agreement between the model's prediction of the states (tiger door, $\widehat{TD}_t$; grid, $\widehat{GR}_t$) and the actual states reported by the participants ($TD_t^*$ and $GR_t^*$, respectively) as their predictions. The two types of states were predicted by

$$\widehat{TD}_t = \text{argmax}_{s_{TD}} P_{t+1}(s_{TD,t+1}|d_{t+1}) \tag{15}$$

$$\widehat{GR}_t = \text{argmax}_{s_{GR}} P_{t+1}(s_{GR,t+1}) \tag{16}$$

When Eqs. (15) and (16) yielded multiple equally probable states (i.e., maximum a posteriori [MAP]), we assumed that the model randomly extracted one of the MAP states as its prediction.

## Image acquisition and analysis

A 3.0-Tesla Siemens MAGNETOM Prisma fit scanner (Siemens Healthineers, Erlangen, Germany) with a standard 64-channel phased-array head coil was used for image acquisition. We acquired interleaved T2*-weighted echo-planar images (TR, 1000 ms; TE, 30 ms; flip angle, 50°; matrix size, 100 × 100; field of view, 200 × 200; voxel size, 2 × 2 × 2.5 mm; and number of slices, 66). Volume acquisition was synchronised with the onset of each delay period. We also acquired whole-brain high-resolution T1-weighted structural images using a standard MPRAGE sequence (TR, 2250 ms; TE, 3.06 ms; flip angle, 9°; field of view, 256 × 256; voxel size, 1 × 1 ×1 mm).

Imaging data were analysed using SPM12 (Wellcome Department of Cognitive Neurology, London, UK). For each participant, all functional images were preprocessed, including slice-timing correction, spatial realignment, co-registration to the individual high-resolution anatomical image, normalisation to an MNI template, and smoothing with a Gaussian kernel filter (FWHM, 8 mm). In addition, high-pass filtering with a cut-off of 128 s was applied to remove low-frequency drifts from the signal.

Our fMRI analyses were conducted using standard GLMs, which employ event-related regressors convolved with the canonical hemodynamic response function. The basic first-level design matrix of the GLMs included a constant term, 6 motion parameters as nuisance regressors, and 12 event-related regressors in each trial (events indicated by numbers in Fig. 1a) per session. The four regressors for the delay periods before the prediction and confidence report (corresponding to steps 4, 6, 8, and 11 in Fig. 1a) were modelled as boxcar functions with a duration of 4 s, whereas the other regressors were modelled as delta functions. The regression analyses were conducted for the following four GLMs, each of which had additional specific regressors of interest in the model (see below).

GLM1: The action feedback period (Step 3 in Fig. 1a) was modelled as two independent events according to the action selected by the participant (Listen and Move) to identify the brain regions involved in feedback-based processing based on the types of actions (Fig. 4a, Supplementary Fig. 3).

GLM2: The action feedback period in the Listen trial (step 3a in Fig. 1a) was modelled as two independent modes of information processing based on our hierarchical inference model (re-estimation and updating modes) to identify the brain regions in the different inference procedures after the same action (Fig. 4b, c).

GLM3: For the tiger door position and grid location prediction events (steps 4 and 8 in Fig. 1a), the participant's subjective confidence level in each prediction (0.5 for high confidence and −0.5 for low confidence) was introduced as a parametric modulator to examine confidence-related brain activity (Fig. 5, cool colour scale). Each prediction onset was modulated only by the corresponding confidence level.

GLM4: For the tiger door position and grid location predictions (steps 4 and 8 in Fig. 1a), the entropy of each inference, calculated based on the computational model (hierarchical inference model, Eqs. (17) and (18)), was introduced as a parametric modulator (Fig. 5, warm colour scale). The

variables were scaled to the range [0,1] and mean-centred (that is, orthogonalized for the constant term).

$$U_{TD,t} = -\sum_{s_{TD}} P_t(s_{TD}) \log P_t(s_{TD}) \tag{17}$$

$$U_{GR,t} = -\sum_{s_{GR}} P_t(s_{GR}) \log P_t(s_{GR}) \tag{18}$$

A random-effects analysis was performed at the group level using an anatomically localised cerebral cortex. Statistical thresholds were set at the voxel level of $p < 0.001$ (uncorrected) and the cluster level of $p < 0.05$ (FWE-corrected).

## Statistics and reproducibility
We analysed data from $N = 20$ healthy human participants. The task was implemented using Psychopy3[90]. The behavioural analyses were performed using MATLAB R2023a (Mathworks, Natick, Massachusetts, US) and R Statistical Software version 4.2.3[91]. The imaging analyses were performed using SPM12.

## Reporting summary
Further information on research design is available in the Nature Portfolio Reporting Summary linked to this article.

## Data availability
Anonymized behavioural data are available on GitHub (https://github.com/RisaKatayama/article-belief-hierarchical-spatial.git). Unthresholded group-level statistical maps are available on NeuroVault (https://neurovault.org/collections/NZJMDMFQ/).

## Code availability
All codes supporting our main results presented in this study are available on GitHub (https://github.com/RisaKatayama/article-belief-hierarchical-spatial.git).

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

## Acknowledgements
This study was supported by a project (No. P20006) subsidised by the New Energy and Industrial Technology Development Organization (NEDO) and by JSPS KAKENHI (No. 22H04998 and 23H04676), Japan. W.Y. was funded by MRC/Versus Arthritis (MR/W027593/1) and the Wellcome Trust (203139/Z/ 16/Z and 203139/A/16/Z), and the NIHR Oxford Health Biomedical Research Centre (views expressed are those of the authors and not necessarily those of the NIHR or the Department of Health and Social Care). R.K. was funded by JST, the establishment of a university fellowship toward the creation of science technology innovation (Grant Number JPMJFS2123), Japan. The funding agencies had no role in the study design, data collection and analysis, decision to publish or preparation of the manuscript. The authors thank B. Seymour for invaluable comments to improve this study.

## Author contributions
S.I. conceived the project; R.K., R.S., S.I. and W.Y. designed the research; R.S. performed the experiments; R.K. analysed the data; R.K. wrote the draft; and R.K., S.I. and W.Y. prepared the final manuscript.

## Competing interests
The authors declare no competing interests.
