## [Peer review file · Communications Biology]

Reviewers' comments:

Reviewer #1 (Remarks to the Author):

The paper introduces a novel behavioral paradigm to study neural computations associated with hierarchical inference in the context of spatial navigation. The task requires humans to estimate and report hidden states at two different levels of abstraction in a grid world with a fairly complex transition structure, as well as the confidence associated with those estimates. The authors find that human behavior is consistent with a model in which participants make use of the true generative model and perform hierarchical inference. The authors also find that activity in posterior/anterior regions of dmPFC and putamen are more correlated with inference of less/more abstract hidden states.

Overall, I think this is a really important study and I would like to see it published. One of the key strengths is the novelty of the paradigm. The study introduces a task that allows for investigating hierarchical inference in a naturalistic setting without sacrificing structural constraints, making it amenable to computational analysis. The vast majority of studies investigating this computation tend to use extremely simplified tasks with unclear relevance to real-world problems, so this study as an important step towards understanding complex computations in real-world settings. However some basic details are either missing/confusing which, if addressed, will make this work more accessible and interesting to readers. I list them below.

(1) The first paragraph mentions the mean accuracy of behavioral predictions for Tiger door (TD) and Grid location (GL). I'm curious what these accuracies are when listen and move trials are analyzed separately. I would imagine that the accuracy of GL prediction should not depend on the type of action but TD prediction would. Is that true?

(2) How does the accuracy of the predictions evolve as a function of trials? Figure 2 shows how the proportion of listen actions and confidence evolve during the game but a similar plot for accuracy should be included here.

(3) Do all participants listen in the first 2 trials and the move in the third trial? If so, it is important to mention this in the text along with an explanation for the dip in panel 2a.

(4) Line 142 attributes a "lack of correlation" to different processes for computing TD and GL predictions but the reported correlation value of 0.47 is pretty high. So either the conclusion should change or the authors must provide more context to understand why this level of correlation is considered to be low.

(5) The majority of p-values are 8.9×10^{-5} . Either there is a typo or the statistics being compared in different statistical tests share something in common. This begs an explanation.

(6) Line 167 mentions the accuracy of the best model in the predicting participants' responses. I have a few questions about this:

(6.1) How were the model predictions when listen and move trials are analyzed separately?

(6.2) How does this compare to the predictions of other models that were tested. Supplementary table 1 shows BIC values. Those are helpful but the main text should include statistics or a figure showing the prediction accuracy of all model variants. Without that, it is difficult to appreciate the central claims of the paper about how humans perform this task.

(6.3) What are the chance levels? Chance for TD and GL would be $1/3$ and $1/16$, is that correct? Why does the model perform so well in predicting participants' GL decision (93%)? This is higher than even the TD decision (88%). The text should include an intuitive explanation for this.

(7) Is the higher activity during listening than moving simply due to differences in the nature of visual feedback rather than cognitive factors?

(8) The analysis of imaging data is rather simple and does not make use of the model, except for entropy which reveals brain networks that are already seen in confidence reports. This is fine, but would be good to at least discuss possible ways to make use of the model parameters to interpret neural activity. Otherwise, the behavioral and imaging sections of the paper would seem somewhat disconnected.

(9) About the model:

(9.1) I suppose the distinction between the computations for updating vs re-estimating comes about because the authors use a MAP estimate for the current state. Is that correct? If computations are fully probabilistic i.e., if uncertainty is propagated through the update step, then a distinct re-estimation step will not be necessary. Since a large portion of the analysis makes use of this distinction, the authors must motivate the reason for introducing them as distinct computations in the text e.g., whether this is because fitting the POMDP model is otherwise intractable and why.

(9.2) There are 3 candidate tiger doors (L, R, F) but only two possible observations (roar or not) so it took me a while to understand the meaning of $s=v$ in equation 2 of the methods. v should be

redefined as the location of the roaring door rather than "observation of roars" in the methods to avoid confusion.

(9.3) The specific form of the three cases for update (equation 3) and re-estimation (equation 4) need to be explained. They seem reasonable but I don't exactly understand how they were derived. It would be extremely helpful to provide a derivation of these equations starting from the inference equations of the POMDP and explicitly introducing the assumptions that led to the derivation of equations 3 and 4, particularly how β and γ come about. This would be of great help to readers who wish to adopt this mathematical framework. The choice of introducing δ in equation 2 also needs to be motivated. In general, it seems that the authors assume that humans use some heuristics to perform approximate inference over the POMDP. This is totally fine as long as the heuristics are spelled out explicitly at least in the methods, if not in the main text.

(9.4) The model parameters are hidden away in a supplementary table. The parameter values would be of great interest to readers since they are very informative about the computational strategy used by humans. I'd like to see this included in the main text along with an interpretation of the best fit parameter values. The impact of this study will be much higher if the authors put some effort into interpreting the parameters of the computational model rather than just focusing on the structure of the model which is quite generic.

(10) My understanding is this study rewards participants for exploring more states while typical navigation paradigms reward participants for reaching a particular goal state. Was the reward function (equation 1) taken into account when modeling human behavior? If not, it would be useful to discuss how that could be done.

(11) There are some missing words below equation 2: "where is the probability of roar observations". I guess it the authors are referring to α ?

(12) Typo in the x-label of panels in 3c. Both panels are labeled as "listen" but one of them should be "move".

(13) The discussion would benefit from contrasting this paradigm with other works on human spatial navigation in mazes: DA Simon & N Daw (2011), S Zhu et al. (2022), W de Cothi et al. (2022) etc.

(14) I appreciate the authors for making the data and code freely available. It would be more ideal if the data are annotated so readers know what the different variable names mean.

Overall, I think this is a really creative study and I congratulate the authors for attempting something new and potentially significant.

Reviewer #2 (Remarks to the Author):

This study characterizes two distinct sources of uncertainty in the navigation problem, which is usually formulated as a partially observable Markov decision process (POMDP). One is observational uncertainty, and the other is state uncertainty. The authors designed the Tiger maze experiment and used hierarchical models to quantify both types of uncertainty. They also analyzed fMRI data to locate different brain regions of such information.

Overall, I enjoyed reading the manuscript. I think the experiment is well designed, and the modeling of hierarchical POMDP is both important and novel in neuroscience. I found the derivation of state transition matrix from behavioral data and its interpretation of the order of inference interesting.

Perhaps my main concern is that the analysis is still largely descriptive only, also with very explicit choices of the model architecture. If the experimenter can show some causal interventions based on the learned model, it is more evident that the model faithfully describes the actual cognitive processes. In addition, if the hierarchy was not imposed but purely learned from data, it would be more impressive.

I listed a few questions for the authors below.

Line 144: You mentioned "different time scales", can you elaborate on that? Do you mean they happen in a particular order, or do you mean they happen at different speeds?

Line 149: The behavioral data suggests that "the grid was hierarchically inferred after the tiger door inference", did you find such evidence in the brain activity? E.g. within one trial?

Fig 2: "Trial number" was used in Fig 2a, 2c, I find it a little confusing. Is the total number of trials of a game, or the index of a trial within a game? In other words, does trial 4 mean a game prematurely ended with 4 trials, or the 4th trial of any game? And in Fig 2c, why are "listen trials" and "move trials" at most 6 trials?

Fig 2a: Why is there a drop at trial 3? Is this a stereotypical behavior of all subjects?

Reviewer #3 (Remarks to the Author):

Thank you for the invitation to review "Belief inference for hierarchical hidden states in spatial navigation" by Katayama et al. This study used a POMDP approach to computationally model a maze

navigation task with an elegantly designed variation on the Tiger problem. The authors found several neural correlates for the POMDP parameters.

I enjoyed reading this paper and thought that it was very well-written paper and describes a well-designed experiment. I feel that this work contributes interesting findings at the intersection of navigation, inference, and metacognition which I believe will be of interest to a computational cognitive neuroscience audience. I was particularly struck by the behavioral and neural results linking confidence and the entropy of inferences.

In my opinion, a strength of this paper was the behavioral modeling and the experimental design. As such, I would have liked to see more discussion about that and substantially less about the brain regions of interest and their purported functions, which could possibly be moved to supplementary discussion if there are space limitations. I would appreciate it if the authors could address in their discussion:

1. Consideration of other relevant navigation models or why they are ruled out (perhaps because they are too simplistic) in modeling this task
2. Relationship to the reinforcement learning / approach-avoid literature
3. Information theory interpretations of the Bayesian inference framework
4. Metacognition literature on confidence ratings, what you would expect to happen to neural activity, how that activity could represent probability distributions.
5. Implications for these findings to generalize to understanding similar forms of behavior or directions for future study

Minor comments/questions:

1. I do not believe it is useful to visualize both uncorrected and thresholded p in the same brain map in Fig 4b. Given that the uncorrected maps were uploaded to Neurovault, the authors could simply refer the readers there and only visualize the thresholded p map.
2. Could the authors report the spatial correlation between the confidence and entropy maps in 5a-b?
3. Please add a paragraph on strengths and limitations of the study, e.g. modest sample size, limitations of model fitting, consideration of cross-validation of fits, etc.

Brown: Reviewer's comments

Black: Our replies

Blue: Extracts from the manuscripts

Responses for Reviewer 1:

Overall, I think this is a really important study and I would like to see it published. One of the key strengths is the novelty of the paradigm. The study introduces a task that allows for investigating hierarchical inference in a naturalistic setting without sacrificing structural constraints, making it amenable to computational analysis. The vast majority of studies investigating this computation tend to use extremely simplified tasks with unclear relevance to real-world problems, so this study as an important step towards understanding complex computations in real-world settings. However some basic details are either missing/confusing which, if addressed, will make this work more accessible and interesting to readers. I list them below.

- (1) The first paragraph mentions the mean accuracy of behavioral predictions for Tiger door (TD) and Grid location (GL). I'm curious what these accuracies are when listen and move trials are analyzed separately. I would imagine that the accuracy of GL prediction should not depend on the type of action but TD prediction would. Is that true?

This is a very interesting point. The following table summarises the prediction accuracy of the participants and also the prediction accuracy of the model (in response to comment #6.1) for each type of behaviour.

Behavioural accuracy	Listen	Move	Wilcoxon signed-rank test
Tiger door position	84.1±4.2%	76.8±4.8%	$p=2.2\times 10^{-4}$
Grid location	45.0±9.0%	67.8±8.8%	$p=8.9\times 10^{-5}$
Model accuracy			
Tiger door position	94.1±3.4%	80.0±4.0%	$p=8.9\times 10^{-54}$
Grid location	49.4±5.3%	72.3±7.3%	$p=8.9\times 10^{-5}$

Although significant differences were found for all items, it is not appropriate to simply compare prediction accuracy with action type alone, as the action type was strongly associated with the trial index (Fig. 2a in the main text), i.e., Listen tends to be more distributed early in the game, Therefore, we performed two-way repeated ANOVAs to

dissociate the effect of the trial index and the action type within the participants. As expected by the reviewer, both the trial index ($F(5,14)=24.9$, $p=1.7\times 10^{-6}$) and the action type ($F(1,11)=47.7$, $p=2.6\times 10^{-5}$), for showed significant effects on TD prediction accuracy, whereas for GL prediction accuracy the effect of the trial index was significant ($F(5,14)=36.7$, $p=1.5\times 10^{-7}$) but no effect of the action type was found ($F(1,11)=3.0$, $p=0.10$). This suggests that the resolution of uncertainty of two types of hidden states involved different processes depending on the selected action, and the following result was added to the revised manuscript (please refer to the reply for the comment #4 of Reviewer 1). Here, the data from the 1st, 2nd and 12th trials were excluded from the analyses because the mean number of trials in which one of two types of action was chosen was extremely low (less than three trials).

In the Results section (line 153):

“The prediction accuracies exhibited the similar temporal profiles as the confidence levels for these two types of predictions (Fig. 2d; repeated two-way ANOVA, for the tiger door, effect of the trial index, $F(5,14)=24.9$, $p=1.7\times 10^{-6}$; effect of the action type, $F(1,11)=47.7$, $p=2.6\times 10^{-5}$; for the grid, effect of the trial index, $F(5,14)=36.7$, $p=1.5\times 10^{-7}$; effect of the action type, $F(1,11)=3.0$, $p=0.11$).”

- (2) How does the accuracy of the predictions evolve as a function of trials? Figure 2 shows how the proportion of listen actions and confidence evolve during the game but a similar plot for accuracy should be included here.

Thank you for your comment. The temporal variation in prediction accuracy showed a similar profile to the confidence level for both the tiger door position and grid location. Previously, the results were shown in Supplementary Figure (2d, i); in the new manuscript, we have added the results to Figure 2 and added the following explanatory text in the manuscript (in the Results section, line 153):

“The prediction accuracies exhibited the similar temporal profiles as the confidence levels for these two types of predictions (Fig. 2d; repeated two-way ANOVA, for the tiger door, effect of the trial index, $F(5,14)=24.9$, $p=1.7\times 10^{-6}$; effect of the action type, $F(1,11)=47.7$, $p=2.6\times 10^{-5}$; for the grid, effect of the trial index, $F(5,14)=36.7$, $p=1.5\times 10^{-7}$; effect of the action type, $F(1,11)=3.0$, $p=0.11$).”

- (3) Do all participants listen in the first 2 trials and the move in the third trial? If so, it is important to mention this in the text along with an explanation for the dip in panel 2a.

Thank you for pointing this out. The first move action was selected in the third trial in $64.2 \pm 22.2\%$ of the games. This concentration within a specific trial index is notably higher than other Move trials, hence the significant dip in the figure.

We have now added this explanation as follows (in the Results section, line 129):

"Note that the significant decrease in the proportion of the Listen trials in the third trial is due to the fact that the participants typically selected their first move action in this specific trial index ($64.2 \pm 22.2\%$ of all games)."

- (4) Line 142 attributes a "lack of correlation" to different processes for computing TD and GL predictions but the reported correlation value of 0.47 is pretty high. So either the conclusion should change or the authors must provide more context to understand why this level of correlation is considered to be low.

Thank you for pointing this out and we acknowledge that the previous data was insufficient to support our conclusion.

The confidence for the tiger door prediction and the grid prediction increased as the trial indices increased, but showed different trends; in the Listen trials, the correlation between the averaged confidence level for the tiger door prediction and the trial index was significantly weaker ($r=0.35 \pm 0.43$) than the correlation of the grid confidence level ($r=0.77 \pm 0.27$; Wilcoxon signed rank test, $p=1.9 \times 10^{-3}$), whereas in the Move trials, both the tiger door and grid prediction confidence had a strong correlation with the trial index, with no significant difference (tiger door, $r=0.81 \pm 0.09$; grid, $r=0.83 \pm 0.09$; $p=0.14$). In addition, the repeated two-way ANOVAs, which we performed to dissociate the effect of the trial index and the action type (please also refer to the reply for the comment #1 of Reviewer 1), showed significant effects on the tiger door confidence for both the trial index ($F(5,14)=25.0$, $p=1.6 \times 10^{-6}$) and the action type ($F(1,11)=72.1$, $p=3.7 \times 10^{-6}$), whereas for the grid confidence the effect of the trial index was significant ($F(5,14)=40.7$, $p=7.5 \times 10^{-8}$) but the effect of the action type was not ($F(1,11)=3.0$, $p=0.11$). These results suggested that the confidence for these two types of prediction involved different information processing depending on the selection actions.

Now we modified the corresponding part as follows (in the Results section, line 145):

"As the Listen and Move trial indices increased, confidence in predicting both the tiger door and the grid location increased, but with different trends. For the Listen trials, confidence in predicting the tiger door increased rapidly even early in the task (Fig. 2c left), and the correlation between the mean confidence level and the trial index was weak ($r=0.35\pm0.43$), whereas for the grid location prediction, confidence increased gradually, i.e., the correlation was strong ($r=0.77\pm0.27$); there was a significant difference in correlation ($p=1.9\times10^{-3}$). On the other hand, when the Move was chosen, both the tiger door and grid predictions had a strong correlation between their confidence level and the trial index (tiger door: $r=0.81\pm0.09$, grid: $r=0.83\pm0.09$), with no significant difference in correlation ($p=0.14$). The repeated two-way ANOVAs also showed significant effects on the tiger door confidence for both the trial index ($F(5,14)=25.0$, $p=1.6\times10^{-6}$) and the action type ($F(1,11)=72.1$, $p=3.7\times10^{-6}$), whereas for the grid confidence the effect of the trial index only was significant ($F(5,14)=40.7$, $p=7.5\times10^{-8}$; effect of the action type, $F(1,11)=3.0$, $p=0.11$). The prediction accuracies exhibited the similar temporal profiles as the confidence levels for these two types of predictions (Fig. 2d; repeated two-way ANOVA, for the tiger door, effect of the trial index, $F(5,14)=24.9$, $p=1.7\times10^{-6}$; effect of the action type, $F(1,11)=47.7$, $p=2.6\times10^{-5}$; for the grid, effect of the trial index, $F(5,14)=36.7$, $p=1.5\times10^{-7}$; effect of the action type, $F(1,11)=3.0$, $p=0.11$). The difference in the temporal profiles of the confidence level evolution suggests that these two predictions involve different processes depending on the selected actions."

- (5) The majority of p-values are 8.9×10^{-5} . Either there is a typo or the statistics being compared in different statistical tests share something in common. This begs an explanation.

Thank you for pointing this out.

When the signs of all data samples are positive for the null hypothesis, i.e., all differences between paired samples are greater than zero, the Z-value in the Wilcoxon signed-rank test is a lower bound that depends only on the sample size n . In other ward, for $n=20$ in our case, the value is $p=8.8574\dots\times10^{-5}\doteq8.9\times10^{-5}$.

The following shows how the Z-value is calculated in the Wilcoxon signed-rank test.

$$T = \min(W^+, W^-)$$
$$Z = \frac{\left| T - \frac{n(n+1)}{4} \right|}{\sqrt{\frac{n(n+1)(2n+1)}{24}}}$$

where $W^+ = \sum_{d_i > 0} R_i$ and $W^- = \sum_{d_i < 0} R_i$

Here d_i is the difference of the paired sample (x_i, y_i) , R_i is the rank of $|d_i|$.

In the scenario where the signs of all data samples are positive, T equals 0, and the Z value is as follows where $n = 20$:

$$Z = \frac{\frac{n(n+1)}{4}}{\sqrt{\frac{n(n+1)(2n+1)}{24}}} = 3.9133$$

so that the corresponding p value (two-sided Wilcoxon signed rank test) is $p = 8.8574 \dots \times 10^{-5} \approx 8.9 \times 10^{-5}$.

(6) Line 167 mentions the accuracy of the best model in the predicting participants' responses.

I have a few questions about this:

(6.1) How were the model predictions when listen and move trials are analyzed separately?

Thank you for this comment. The model's prediction accuracies were affected by the action type (please see to the response for the comment #1), however, this difference was partially due to the strong relationship between the action type and the trial index. We performed a two-way repeated ANOVA to examine the effect of the action type, taking into account the trial index, on the prediction accuracy of the model. The results showed that both the trial index and the action type had a significant main effect on the prediction accuracy on the tiger door choices (trial, $F(5,14) = 61.2$, $p = 5.2 \times 10^{-9}$; action, $F(1,11) = 2.1 \times 10^2$, $p = 1.5 \times 10^{-8}$), whereas only the trial index had a main effect on the reproducibility on the grid prediction (trial, $F(5,14) = 96.4$, $p = 2.5 \times 10^{-10}$; action, $F(1,11) = 2.5 \times 10^{-2}$, $p = 0.88$). The results of this analysis, together with the response to comment #1, suggest that the prediction accuracy of the model improves as the participant's behaviour becomes more correct; this finding strengthens the validity of our model.

(6.2) How does this compare to the predictions of other models that were tested. Supplementary table 1 shows BIC values. Those are helpful but the main text should include statistics or a figure showing the prediction accuracy of all model variants. Without that, it is difficult to appreciate the central claims of the paper about how humans perform this task.

Thank you. We have included the model selection statistics as Table 1.

(6.3) What are the chance levels? Chance for TD and GL would be 1/3 and 1/16, is that correct? Why does the model perform so well in predicting participants' GL decision (93%)?

This is higher than even the TD decision (88%). The text should include an intuitive explanation for this.

We appreciate the clarification. Indeed, the chance level of predicting the tiger door position is 1/3, and for grid location is 1/16.

In the previous manuscript, predictions were defined as correct if there was more than one MAP state in the model, and one of which was the state reported by the participant. This meant that the reproducibility of the model could have been high, especially for the grid location prediction. Here, we re-evaluated the reproducibility by assuming that the model randomly extracted one of several MAP states as its prediction. This new, more appropriate evaluation showed that the reproducibility of the grid location prediction was lower (60.9±6.2%) than that of the tiger door prediction (87.2±2.3%), as the reviewer had suggested.

In the revised manuscript, the definition of the prediction accuracy of the model is described in the Methods section and the results have been replaced:

In the Method section (line 607):

“In validating our proposed hierarchical model, we also used the agreement between the model’s prediction of the states (tiger door, \widehat{TD}_t ; grid, \widehat{GR}_t) and the actual states reported by the participants (TD_t^ and GR_t^* , respectively) as their predictions. The two types of states were predicted by*

$$\widehat{TD}_t = \operatorname{argmax}_{s_{TD}} P_{t+1}(s_{TD,t+1}|d_{t+1}) \quad \dots (15)$$

$$\widehat{GR}_t = \operatorname{argmax}_{s_{GR}} P_{t+1}(s_{GR,t+1}) \quad \dots (16)$$

When equations (15) and (16) yielded multiple equally-probable states (i.e., maximum a posteriori [MAP]), we assumed that the model randomly extracted one of the MAP states as its prediction.”

In the Results section (line 193):

“Our hierarchical inference model accurately reproduced the participants’ decisions for both predicting the tiger door position (87.2±2.3%) and predicting the grid location (60.9±6.2%). The reproducibility was significantly better than chance, even in trials in which the

participants' predictions were incorrect (Wilcoxon signed-rank test; tiger door, $56.5 \pm 6.9\%$, $p = 8.9 \times 10^{-5}$; grid, $33.2 \pm 5.4\%$, $p = 8.9 \times 10^{-5}$).

(7) Is the higher activity during listening than moving simply due to differences in the nature of visual feedback rather than cognitive factors?

As the reviewer pointed out, the characteristics of the visual stimuli presented as feedback in the two trials (i.e., information about the tiger door or frame-by-frame images showing movement) are different. However, as the visual stimuli after choosing Listen induced an updating of the inference rather than a simple state transition, we do not believe that brain activity is exclusively dependent on the type of visual stimuli.

In order to extract only the effects of cognitive load, and independent of the differences in the visual stimuli, we compared the brain activity during the feedback period after the first Listen trial in the games and all subsequent Listen trials. The results from the one-way ANOVA indicated significant activation in the bilateral visual cortices, parahippocampal region, and the right rostrolateral prefrontal cortex on the first Listen trials. This activation pattern was consistent with the observations when comparing the Listen and Move trials, and supports that these areas are associated not only with visual stimuli but also with different levels of uncertainty; i.e., cognitive load.

Difference in brain activation between the Listen and Move trials.

(a) Brain areas that exhibited the effect of the index of Listen trials (first or later) on the difference in the brain activity between the Listen and Move trial. The clusters are significant at $p < 0.05$ family-wise error (FWE)-corrected, with the cluster defining threshold $p < 0.0001$ uncorrected.

(b) Overlap between brain regions whose activity was higher in the whole Listen trial compared to the Move trials (green, same as Fig. 4a in our manuscript) and where the activity difference was affected by the index of the Listen trial index (yellow, same as Fig (a)). This figure was not included in the current manuscript.

(8) The analysis of imaging data is rather simple and does not make use of the model, except for entropy which reveals brain networks that are already seen in confidence reports. This is fine, but would be good to at least discuss possible ways to make use of the model parameters to interpret neural activity. Otherwise, the behavioral and imaging sections of the paper would seem somewhat disconnected.

Thank you for pointing this out. In the revised manuscript, we added the discussion about the possible way to utilise the model parameters for the imaging analyses as follows (please also refer to the minor comment #3 of Reviewer 3; in the Discussion section, line 410).

“In our model, the parameters were involved in modifying the mechanisms that integrate observation and the two types of inference, and may control the functional connectivity between these neural substrates. In this study, however, these hypotheses were not assessed mainly due to the limitations of the temporal resolution of the fMRI signals and the event-related task design. An intriguing avenue for future research lies in exploring the hierarchical functional networks within the brain and their correspondence with the computational models, potentially by employing higher temporal resolution measurements and analytical methods, and by using intervention-based approaches to verify causality in the brain network.”

(9) About the model:

(9.1) I suppose the distinction between the computations for updating vs re-estimating comes about because the authors use a MAP estimate for the current state. Is that correct? If computations are fully probabilistic i.e., if uncertainty is propagated through the update step, then a distinct re-estimation step will not be necessary. Since a large portion of the analysis makes use of this distinction, the authors must motivate the reason for introducing them as distinct computations in the text e.g., whether this is because fitting the POMDP model is otherwise intractable and why.

We appreciate this important comment.

Regarding the state inference by the model, a MAP estimate is used for the TD inference after the Listen action, which is then used as an observation for the GL inference. This method was introduced by the motivation that the TD observation information (the direction of the door from which the tiger roars) is inherently probabilistic, and as there is only one true TD state, it is appropriate to use the most probable state as the inferred state. In addition, there was a concern that making the TD inference fully probabilistic would increase the state space of the GL inference by a factor of three, increasing the cognitive load to a point that might be difficult for humans to handle with. On the other hand, for the GL inference, MAP estimation was not used after that Move action, as there would be several states that matched the observations (MAP estimate of the TD state), even if they were not the true state.

To evaluate the assumptions of our mode, we also implemented a model in which the TD inference was updated in a fully probabilistic manner, but it did not reproduce the behaviour well (see table below).

Table: For each model, the table summarises its fitting performances (BIC, Bayesian information criteria; MF, model expected probability; XP, model exceedance probability).

Model	BIC	MF	XP
Hierarchical inference with re-estimation	316.8±76.7	0.65	0.997
Hierarchical inference without re-estimation	341.6±80.3	0.15	2.1×10^{-3}
Fully-probabilistic hierarchical inference	573.8±38.5	0.040	0
Parallel inference	399.5±57.1	0.040	0
Top-down inference	343.4±79.7	0.12	9.0×10^{-4}

We have added the motivation for introducing the MAP estimation method for the TD inference in the Methods in the revised manuscript as follows (line 567):

“The grid location was inferred using a method that depended on whether the direction of the observed tiger roar matched their prediction or not. Here, the MAP estimate (a single most likely direction) was used for the tiger door position, rather than the fully probabilistic estimate. This is because the TD observation information (the direction of the door in which the tiger roars) is inherently probabilistic, and there is only one true tiger door state.”

(9.2) There are 3 candidate tiger doors (L, R, F) but only two possible observations (roar or not) so it took me a while to understand the meaning of $s=v$ in equation 2 of the methods. v should be redefined as the location of the roaring door rather than "observation of roars" in the methods to avoid confusion.

Thanks. We defined v as "the position of the roaring door" in the revised manuscript.

(9.3) The specific form of the three cases for update (equation 3) and re-estimation (equation 4) need to be explained. They seem reasonable but I don't exactly understand how they were derived. It would be extremely helpful to provide a derivation of these equations starting from the inference equations of the POMDP and explicitly introducing the assumptions that led to the derivation of equations 3 and 4, particularly how β and γ come about. This would be of great help to readers who wish to adopt this mathematical framework. The choice of introducing δ in equation 2 also needs to be motivated. In general, it seems that the authors assume that humans use some heuristics to perform approximate inference over the POMDP. This is totally fine as long as the heuristics are spelled out explicitly at least in the methods, if not in the main text.

Thank you for this comment. We have rewritten the derivation of the formulae for updating and re-estimating of the grid inference, as detailed below (in the Methods section, line 512):

“When the observed position of the roaring door (v_t) agreed with the tiger door prediction (\hat{s}_{TD}), participants credited the history of the observations in the current grid and updated the grid location probabilities. If participants perfectly memorised the maze structure, the grid location probabilities would be updated in the form of Bayesian filtering as follows:

$$P_{t+1}(s_{GR,t+1}|v_{t':t}, \hat{s}_{TD,t+1}) = P_{UD,t+1}(s_{GR,t+1}|v_{t':t}, \hat{s}_{TD,t+1})$$

$$\propto \begin{cases} \beta P_t(s_{GR,t}|v_{t':t}) & \text{if } s_{TD}^*(s_{GR,t}, d_t) = v_t \\ (1 - \beta) P_t(s_{GR,t}|v_{t':t}) & \text{if } s_{TD}^*(s_{GR,t}, d_t) \neq v_t \\ 0 & \text{if } s_{TD}^*(s_{GR,t}, d_t) = \text{back} \end{cases}$$

where $\hat{s}_{TD,t+1} = \operatorname{argmax}_{s_{TD}} P_{t+1}(s_{TD,t+1})$... (3)

s_{TD}^* is the true tiger door position in the grid s_{GR} with an observable body orientation d_t . β is the degree of dependence on the tiger door prediction; if $\beta = 1$, participants extract grids for which $s_{TD}^*(s_{GR}, d_t)$ is consistent with s_{TD} as the candidates and consider all others as unlikely. t' denotes the number of the first trials after transfer to the current grid $s_{GR,t}^*$. Because the door behind participants were always passable, grids where $s_{TD}^*(s_{GR}, d_t)$ was the backside should be excluded from the candidates of the grid location (the third case of Equation (3)). Here, if the participants' memory incomplete, i.e., they erred in recalling the maze structure with the probability γ , they mistakenly update the grid location probability according to the second case in Equation (3), even when the true tiger door position matched the roaring door position (corresponding to the first case), with the error probability γ : similarly, when the true tiger door position was inconsistent to the roaring door position (corresponding to the second case), the update in the first case in Equation (3) could occur with γ . Participants also could update the grid location where $s_{TD}^*(s_{GR}, d_t)$ was backside if they made a mistake about the maze structure (corresponding to the third case). In summary, the dynamics of the grid inference are defined as follows (update mode, step 4a+ in Supplementary Fig. S2a):

$$P_{t+1}(s_{GR,t+1}|v_{t':t}, \hat{s}_{TD,t+1}) = P_{UD,t+1}(s_{GR,t+1}|v_{t':t}, \hat{s}_{TD,t+1}) \\ \propto \begin{cases} [(1-\gamma)\beta + \gamma(1-\beta)]P_t(s_{GR,t}|v_{t':t}) & \text{if } s_{TD}^*(s_{GR,t}, d_t) = v_t \\ [\gamma\beta + (1-\gamma)(1-\beta)]P_t(s_{GR,t}|v_{t':t}) & \text{if } s_{TD}^*(s_{GR,t}, d_t) \neq v_t \\ \gamma P_t(s_{GR,t}|v_{t':t}) & \text{if } s_{TD}^*(s_{GR,t}, d_t) = \text{back} \end{cases} \\ \text{where } \hat{s}_{TD,t+1} = \text{argmax}_{s_{TD}} P_{t+1}(s_{TD,t+1}) \quad \dots (4)$$

In contrast, if v_t was inconsistent with $\hat{s}_{TD,t}$, participants rejected the inference $P_t(s_{GR,t})$ based on the previous observations $v_{t':t}$, and re-estimated under the current observation v_t and the probabilities at trial t' as the prior distribution (re-estimate mode, step 4a in Supplementary Fig. S2a). Here, it is assumed that even in the re-estimation, participants may (with a certain probability ε) continue to update their inferences based on the previous (unreliable) observations. In summary, the dynamics of the grid inference when $\hat{s}_{TD,t} \neq v_t$ are defined as follows:

$$P_{RE,t+1}(s_{GR,t+1}|v_{t':t}, \hat{s}_{TD,t}) = \begin{cases} [(1-\gamma)\beta + \gamma(1-\beta)]P_{t'}(s_{GR,t'}) & \text{if } s_{TD}^*(s_{GR,t}, d_t) = \hat{s}_{TD,t+1} \\ [\gamma\beta + (1-\gamma)(1-\beta)]P_{t'}(s_{GR,t'}) & \text{if } s_{TD}^*(s_{GR,t}, d_t) \neq \hat{s}_{TD,t+1} \\ \gamma P_{t'}(s_{GR,t'}) & \text{if } s_{TD}^*(s_{GR,t}, d_t) = \text{back} \end{cases} \dots (5)$$

$$P_{t+1}(s_{GR,t+1}|v_{t':t}, \hat{s}_{TD,t+1}) = (1-\varepsilon)P_{RE,t+1}(s_{GR,t+1}|v_{t':t}, \hat{s}_{TD,t+1}) + \varepsilon P_{UD,t+1}(s_{GR,t+1}|v_{t':t}, \hat{s}_{TD,t+1}) \dots (6)$$

Note that, Equation (5) derived from Equation (4), in which the prior probability term $P_t(s_{GR,t}|v_{t':t})$ was replaced with $P_{t'}(s_{GR,t'})$."

(9.4) The model parameters are hidden away in a supplementary table. The parameter values would be of great interest to readers since they are very informative about the computational strategy used by humans. I'd like to see this included in the main text along with an interpretation of the best fit parameter values. The impact of this study will be much higher if the authors put some effort into interpreting the parameters of the computational model rather than just focusing on the structure of the model which is quite generic.

Thank you for your important suggestions. We have added a new table (Table1; indicated in the response to comment #6.2 and #9.1) summarising the model parameter estimates in the revised manuscript.

We also added the interpretation of the fitted model parameter values in the main text as follows (in the Results section, line 168):

*“The tiger door was inferred in a Bayesian fashion, i.e., it was updated as the product of the prior information, which is a prediction made in the previous trial, and the newly observed information (the position of the roaring door), weighted by an exponential parameter, delta. The estimated value of this delta parameter (1.8 ± 0.74 ; Table 1) was greater than 1, indicating that the observed information was given more weight. The grid location was then also updated using Bayesian estimation, with the likelihood information being the extracted grid positions, such that the probability of satisfying the condition of matching the predicted tiger door position was weighted by a parameter beta. Here, it was also assumed that if the observed tiger door position disagreed with the prediction, the participant would re-estimate the grid using the current tiger door prediction (re-estimation), but could continue to update the grid position inferred on the previous trial (which did not match the observation) with probability epsilon. The estimated beta value, which indicated the accuracy of grid extraction from the observations, was high (0.97 ± 0.045), while the epsilon value, which indicates the probability of dragging incorrect estimate was low (0.14 ± 0.29); these results suggest that the participants tended to infer the hidden state accurately from the observations. On the other hand, if they moved to an adjacent grid, the next grid location was inferred from the structure of the maze, which in turn predicted the tiger door position (Fig. 3a; for further details, please refer to Supplementary Fig. S2 and the **Methods**, specifically the **Behavioural model**). In both the Listen and Move trials, the grid location had to be estimated from memory of the maze structure, but to account for participants' imperfect memory, an error rate gamma was introduced: the mean of estimated gamma was very small (0.074 ± 0.057) and the individual*

estimates were negatively correlated with the accuracy of the grid location prediction ($r=-0.55$, $p=1.3\times 10^{-2}$)."

(10) My understanding is this study rewards participants for exploring more states while typical navigation paradigms reward participants for reaching a particular goal state. Was the reward function (equation 1) taken into account when modeling human behavior? If not, it would be useful to discuss how that could be done.

Thank you for your valuable comment. We did not model the participants' decision-making behaviours, but used their action as an observed variable for modelling the hidden state inference. The rationale behind this decision is two folds: first, to prevent unnecessarily increasing the complexity of the model beyond what is essential for modelling hidden state inference, which constitutes the primary focus of this study; and second, because behavioural models are likely to exhibit substantial variations among participants in an environment lacking a particular goal state.

We have added a description of the above model setting in the Methods (line 512) and discussed the modelling of reward-based navigation in the Discussion as follows (line 296):

"Although not introduced in our hierarchical inference model to avoid complication, it would be possible to model the navigation behaviour based on the goal of exploring as many unvisited states as possible to avoid opening the tiger door. Specifically, behaviour in our task could be modelled as a hierarchy of two stages: the decision to listen or move and, if moving, the direction in which to move. The former could be formulated as an approach-avoidance conflict model^{33,34}, where the conflict is between the avoidance behaviour of avoiding the tiger door and the exploration behaviour of reaching the goal. The choice of moving direction could then be determined by the objective function of maximising the explored grids in the maze²⁰."

20. Katayama, R., Yoshida, W. & Ishii, S. Confidence modulates the decodability of scene prediction during partially-observable maze exploration in humans. *Commun. Biol.* **5**, 1–14 (2022).

33. Amemori, K. I. & Graybiel, A. M. Localized microstimulation of primate pregenual cingulate cortex induces negative decision-making. *Nat. Neurosci.* **15**, 776–785 (2012).

34. Zorowitz, S. *et al.* The neural basis of approach-avoidance conflict: A model based analysis. *eNeuro* **6**, 1–12 (2019).

(11) There are some missing words below equation 2: "where is the probability of roar observations". I guess it the authors are referring to α ?

Apology. The following explanation has been added (line 531).

"where α is the probability of roar observations (0.85) and ..."

(12) Typo in the x-label of panels in 3c. Both panels are labeled as "listen" but one of them should be "move".

Sorry, the typo has been corrected.

(13) The discussion would benefit from contrasting this paradigm with other works on human spatial navigation in mazes: DA Simon & N Daw (2011), S Zhu et al. (2022), W de Cothi et al. (2022) etc.

Thank you for your suggestions. In the part of the Introduction describing our task, we referred to the other navigation studies and described the paradigm differences as follows.

In the Introduction section (line 97):

"Here, while most maze navigation studies have assumed an environment with an observable state and an explicit goal state and regarded the navigation process as a decision-making problem²⁷⁻²⁹ or the generation of a spatial predictive map^{30,31}, the Tiger maze navigation task is a problem setting focused on hidden state inference."

27. Simon, D. A. & Daw, N. D. Neural correlates of forward planning in a spatial decision task in humans. *J. Neurosci.* **31**, 5526–5539 (2011).

28. de Cothi, W. *et al.* Predictive maps in rats and humans for spatial navigation. *Curr. Biol.* **32**, 3676-3689.e5 (2022).

29. Anggraini, D., Glasauer, S. & Wunderlich, K. Neural signatures of reinforcement learning correlate with strategy adoption during spatial navigation. *Sci. Rep.* **8**, 1–14 (2018).

30. Zhu, S., Lakshminarasimhan, K. J., Arfaei, N. & Angelaki, D. E. Eye movements reveal spatiotemporal dynamics of visually-informed planning in navigation. *Elife* **11**, 1–34 (2022).

31. Epstein, R. & Kanwisher, N. A cortical representation the local visual environment. *Nature* **392**, 598–601 (1998).

(14) I appreciate the authors for making the data and code freely available. It would be more ideal if the data are annotated so readers know what the different variable names mean.

Thanks. We have added annotations to the variable names on Github.

Brown: Reviewer's comments

Black: Our replies

Blue: Extracts from the manuscripts

Responses for Reviewer 2:

Overall, I enjoyed reading the manuscript. I think the experiment is well designed, and the modeling of hierarchical POMDP is both important and novel in neuroscience. I found the derivation of state transition matrix from behavioral data and its interpretation of the order of inference interesting.

Perhaps my main concern is that the analysis is still largely descriptive only, also with very explicit choices of the model architecture. If the experimenter can show some causal interventions based on the learned model, it is more evident that the model faithfully describes the actual cognitive processes. In addition, if the hierarchy was not imposed but purely learned from data, it would be more impressive.

Thank you for this comment.

As the two hidden states were explicitly separated in the Tiger Maze task, our proposed model also incorporates an explicit hierarchy in the state space. However, as the reviewer points out, we agree that further studies are needed to understand the hierarchical information processing mechanisms that are thought to be innate in the brain, including the construction of computational models that automatically construct the hierarchy of the environment from observations, and interventional experiments to validate these models.

These points are now discussed in a new paragraph added to the Discussion as the limitations of the present study and directions for future research (please refer to the reply for the minor comment #3 of Reviewer 3).

1. Line 144: You mentioned "different time scales", can you elaborate on that? Do you mean they happen in a particular order, or do you mean they happen at different speeds?

We apologise for the misleading wording. We intended to convey that the processing of the two types of predictions vary depending on the participants' selected action (i.e., Listen or Move), but not to refer to the order of information processing; we considered that it was a leap in logic to conclude about the processing order of them based on the correlation analyses only. We have now revised the relevant section (please refer to the reply for the comment #4 of Reviewer 1).

2. Line 149: The behavioral data suggests that “the grid was hierarchically inferred after the tiger door inference”, did you find such evidence in the brain activity? E.g. within one trial?

We appreciate this is indeed an interesting point, but unfortunately, due to the limited temporal resolution inherent to fMRI, we find it challenging to precisely identify the hierarchical structure of brain activity levels.

We acknowledge this as a crucial issue for our argument and address it as a limitation of our study in the revised manuscript as follows (line 407):

“In addition, conclusive evidence concerning the existence of functional hierarchical information processing among these neural substrates has not been presented. It has also not been demonstrated how the model parameters estimated for each individual are related to their neural activity. In our model, the parameters were involved in modifying the mechanisms that integrate observation and the two types of inference, and may control the functional connectivity between these neural substrates. In this study, however, these hypotheses were not assessed mainly due to the limitations of the temporal resolution of the fMRI signals and the event-related task design. An intriguing avenue for future research lies in exploring the hierarchical functional networks within the brain and their correspondence with the computational models, potentially by employing higher temporal resolution measurements and analytical methods, and by using intervention-based approaches to verify causality in the brain network.”

3. Fig 2: “Trial number” was used in Fig 2a, 2c, I find it a little confusing. Is the total number of trials of a game, or the index of a trial within a game? In other words, does trial 4 mean a game prematurely ended with 4 trials, or the 4th trial of any game? And in Fig 2c, why are “listen trials” and “move trials” at most 6 trials?

We apologise for the confusing axis labels in Figure 2a, c and others. We intended to specify the trial index within a single game, where, for example, “trial 4” in Figure 2a corresponds to the fourth trial of the games. In the revised manuscript, we have changed the axis labels as “Index of Listen (Move) trial in a single game” for clarity.

The average number of trials for Listen and Move trials was 5.0 ± 1.6 (maximum 10) and 4.8 ± 2.1 (maximum 9) trials, respectively. However, in the figures, results up to the sixth

trial are presented as data with a sufficient number of samples (more than 5 trials) to justify testing using the one-sided Wilcoxon signed rank test ($p < 0.05$).

The following text has been added to the figure caption (Fig. 2c, line 993):

“Although the number of the Listen and Move trials varied between games (Listen, maximum 10, mean 5.0 ± 1.6 ; Move, 9, 4.8 ± 2.1), the figures show results up to the sixth trial as data with a sufficient number of samples (more than 5 trials) to justify the test (one-sided Wilcoxon signed rank test, $p < 0.05$).”

4. Fig 2a: Why is there a drop at trial 3? Is this a stereotypical behavior of all subjects?

We apologise for the lack of an explanation of this specific behaviour. To address this comment, along with the concerns raised in comment #3 from Reviewer 1, we made modifications to the relevant section (line 129). Please refer to our response to comment #3 from Reviewer 1 for further clarification.

Brown: Reviewer's comments

Black: Our replies

Blue: Extracts from the manuscripts

Responses for Reviewer 3:

I enjoyed reading this paper and thought that it was very well-written paper and describes a well-designed experiment. I feel that this work contributes interesting findings at the intersection of navigation, inference, and metacognition which I believe will be of interest to a computational cognitive neuroscience audience. I was particularly struck by the behavioral and neural results linking confidence and the entropy of inferences.

In my opinion, a strength of this paper was the behavioral modeling and the experimental design. As such, I would have liked to see more discussion about that and substantially less about the brain regions of interest and their purported functions, which could possibly be moved to supplementary discussion if there are space limitations. I would appreciate it if the authors could address in their discussion:

1. Consideration of other relevant navigation models or why they are ruled out (perhaps because they are too simplistic) in modeling this task
2. Relationship to the reinforcement learning / approach-avoid literature
3. Information theory interpretations of the Bayesian inference framework

Thank you for these suggestions. In order to further the discussion on the differences in task setting and behavioural models from related studies, the relevant parts of the Introduction and Discussion have been rewritten as follows (please also refer to the reply for the comment #10 for Reviewer 1).

In the Introduction section (line 97):

"Here, while most maze navigation studies have assumed an environment with an observable state and an explicit goal state and regarded the navigation process as a decision-making problem²⁷⁻²⁹ or the generation of a spatial predictive map^{30,31}, the Tiger maze navigation task is a problem setting focused on hidden state inference."

In the Discussion section (line 291):

"Most previous studies have dealt with maze environments aimed at reaching a goal state, and have formulated navigation behaviour within the framework of reinforcement learning and goal-directed planning²⁷⁻²⁹. However, our Tiger maze navigation task is a problem setting that focuses on hidden state inference and the goal state is unknown, therefore the

participants' navigation behaviour was used as an observed variable rather than being modelled directly by a generative model. Although not introduced in our hierarchical inference model to avoid complication, it would be possible to model the navigation behaviour based on the goal of exploring as many unvisited states as possible to avoid opening the tiger door. Specifically, behaviour in our task could be modelled as a hierarchy of two stages: the decision to listen or move and, if moving, the direction in which to move. The former could be formulated as an approach-avoidance conflict model^{33,34}, where the conflict is between the avoidance behaviour of avoiding the tiger door and the exploration behaviour of reaching the goal. The choice of moving direction could then be determined by the objective function of maximising the explored grids in the maze²⁰.

20. Katayama, R., Yoshida, W. & Ishii, S. Confidence modulates the decodability of scene prediction during partially-observable maze exploration in humans. *Commun. Biol.* **5**, 1–14 (2022).
 27. Simon, D. A. & Daw, N. D. Neural correlates of forward planning in a spatial decision task in humans. *J. Neurosci.* **31**, 5526–5539 (2011).
 28. de Cothi, W. *et al.* Predictive maps in rats and humans for spatial navigation. *Curr. Biol.* **32**, 3676–3689.e5 (2022).
 29. Anggraini, D., Glasauer, S. & Wunderlich, K. Neural signatures of reinforcement learning correlate with strategy adoption during spatial navigation. *Sci. Rep.* **8**, 1–14 (2018).
 33. Amemori, K. I. & Graybiel, A. M. Localized microstimulation of primate pregenual cingulate cortex induces negative decision-making. *Nat. Neurosci.* **15**, 776–785 (2012).
 34. Zorowitz, S. *et al.* The neural basis of approach-avoidance conflict: A model based analysis. *eNeuro* **6**, 1–12 (2019).
4. Metacognition literature on confidence ratings, what you would expect to happen to neural activity, how that activity could represent probability distributions.

Thank you for this comment. In the Discussion, we have referred to the literature on metacognition and shown that the results of our study (i.e., Fig 5) are consistent with previous findings as follows (in the Discussion section, line 343):

“The dACC and precentral gyrus activity, whose activities are correlated with both uncertainty indices, have also been reported in previous metacognition studies to be brain regions that provide metacognitive control signals (i.e., confidence) in various types of decision-making⁵⁷⁻⁵⁹ and learning¹¹. Subjective confidence levels were ... which can be expressed as a probability

distribution. As previous studies suggested that the posterior probability distribution is represented as the multivoxel activity patterns in the localised brain regions^{60,61}, in these regions the inferences may be encoded across multiple neuronal populations.”

11. Meyniel, F. & Dehaene, S. Brain networks for confidence weighting and hierarchical inference during probabilistic learning. *Proc. Natl. Acad. Sci. U. S. A.* **114**, E3859–E3868 (2017).
57. Su, J., Jia, W. & Wan, X. Task-Specific Neural Representations of Generalizable Metacognitive Control Signals in the Human Dorsal Anterior Cingulate Cortex. *J. Neurosci.* **42**, 1275–1291 (2022).
58. Pereira, M. *et al.* Disentangling the origins of confidence in speeded perceptual judgments through multimodal imaging. *Proc. Natl. Acad. Sci. U. S. A.* **117**, 8382–8390 (2020).
59. Fleming, S. M., Huijgen, J. & Dolan, R. J. Prefrontal contributions to metacognition in perceptual decision making. *J. Neurosci.* **32**, 6117–6125 (2012).
60. Glaser, J. I., Perich, M. G., Ramkumar, P., Miller, L. E. & Kording, K. P. Population coding of conditional probability distributions in dorsal premotor cortex. *Nat. Commun.* **9**, (2018).
61. Chan, S. C. Y., Niv, Y. & Norman, K. A. A probability distribution over latent causes, in the orbitofrontal cortex. *J. Neurosci.* **36**, 7817–7828 (2016).

5. Implications for these findings to generalize to understanding similar forms of behavior or directions for future study

Thank you for this suggestion. In the Discussion, we discussed the applicability of our model to other cognitive problems and the direction of future research.

In the Discussion section (line 287):

“Our hierarchical inference model has the potential to be applied to cognitive problems involving hierarchies, such as transfer learning and social interactive decision-making.”

(line 413):

“An intriguing avenue for future research lies in exploring the hierarchical functional networks within the brain and their correspondence with the computational models, potentially by employing higher temporal resolution measurements and analytical methods, and by using intervention-based approaches to verify causality in the brain network.”

Minor comments/questions:

1. I do not believe it is useful to visualize both uncorrected and thresholded p in the same brain map in Fig 4b. Given that the uncorrected maps were uploaded to Neurovault, the authors could simply refer the readers there and only visualize the thresholded p map.

We appreciate the reviewer's opinion. However, the dorsomedial prefrontal cortex (dmPFC) shown in the uncorrected map is a region that, based on our previous study [1], would a priori be expected to show higher activation exhibited during re-estimating than during updating. In addition, the dmPFC is claimed to have a relevant function in decision-making in conflict situations [2-8], and we considered it to be a region of interest to a wide range of readers. For these reasons, we would like to keep the uncorrected map in Figure 4b.

1. Yoshida, W. & Ishii, S. Resolution of Uncertainty in Prefrontal Cortex. *Neuron* **50**, 781–789 (2006).
 2. Pochon, J. B., Riis, J., Sanfey, A. G., Nystrom, L. E. & Cohen, J. D. Functional imaging of decision conflict. *J. Neurosci.* **28**, 3468–3473 (2008).
 3. Fleming, S. M., Van Der Putten, E. J. & Daw, N. D. Neural mediators of changes of mind about perceptual decisions. *Nat. Neurosci.* **21**, 617–624 (2018).
 4. Fleck, M. S., Daselaar, S. M., Dobbins, I. G. & Cabeza, R. Role of prefrontal and anterior cingulate regions in decision-making processes shared by memory and nonmemory tasks. *Cereb. Cortex* **16**, 1623–1630 (2006).
 5. Heereman, J., Walter, H. & Heekeren, H. R. A task-independent neural representation of subjective certainty in visual perception. *Front. Hum. Neurosci.* **9**, 1–12 (2015).
 6. Botvinick, M. M., Carter, C. S., Braver, T. S., Barch, D. M. & Cohen, J. D. Conflict monitoring and cognitive control. *Psychol. Rev.* **108**, 624–652 (2001).
 7. Boldt, A. & Yeung, N. Shared neural markers of decision confidence and error detection. *J. Neurosci.* **35**, 3478–3484 (2015).
 8. Holroyd, C. B. & Coles, M. G. H. The neural basis of human error processing: Reinforcement learning, dopamine, and the error-related negativity. *Psychol. Rev.* **109**, 679–709 (2002).
2. Could the authors report the spatial correlation between the confidence and entropy maps in 5a-b?
 3. Please add a paragraph on strengths and limitations of the study, e.g. modest sample size, limitations of model fitting, consideration of cross-validation of fits, etc.

Thank you for the comment. The spatial correlation of active regions in terms of the confidence and the entropy was 0.80 and 0.85 for the tiger door inference and the grid location inference, respectively. Here, the Szymkiewicz–Simpson coefficient was used as the correlation measure, as the voxel size of the significant clusters differ significantly (tiger door: confidence, 2,298 voxels; entropy, 11,631 voxels; grid location confidence: 3,865 voxels; entropy, 12,261 voxels).

In the revised manuscript, these results are reported as follows (in the Results section, line 248):

“The brain regions that ... also extended to other areas (Szymkiewicz-Simpson coefficient for the tiger door confidence and entropy, $SS=0.80$; for the grid confidence and entropy, $SS=0.85$; see also Fig. 5a, b, warm colour scale, and Supplementary Table S5).”

3. Please add a paragraph on strengths and limitations of the study, e.g. modest sample size, limitations of model fitting, consideration of cross-validation of fits, etc.

Thank you for this comment. We have now modified the discussion to clarify the strengths and limitations of our study, as follows (in the Discussion section, line 399):

“One notable strength lies in the innovative use of computational modelling, which allowed us to reproduce human hierarchical inference processes as the spatial inference problem in the navigation task, and to extract the neural substrates involved in inference using both subjective (i.e., confidence) and objective measures (uncertainty). However, it is important to acknowledge that cognitive research using models has its limitations. In our case, the computational model was explicitly hierarchical since the task presented an explicit hierarchy of states, whereas in a real-world environment it is natural to assume that humans’ hierarchies or subdivide problems themselves based on the structure of states; this may overlook complex and important mechanisms of cognitive phenomena. In addition, conclusive evidence concerning the existence of functional hierarchical information processing among these neural substrates has not been presented. It has also not been demonstrated how the model parameters estimated for each individual are related to their neural activity. In our model, the parameters were involved in modifying the mechanisms that integrate observation and the two types of inference, and may control the functional connectivity between these neural substrates. In this study, however, these hypotheses were not assessed mainly due to the limitations of the temporal resolution of the fMRI signals and the event-related task design. An intriguing avenue for future research lies in exploring the hierarchical

functional networks within the brain and their correspondence with the computational models, potentially by employing higher temporal resolution measurements and analytical methods, and by using intervention-based approaches to verify causality in the brain network.”

REVIEWERS' COMMENTS:

Reviewer #1 (Remarks to the Author):

The concerns I raised have all been adequately addressed by the authors. I have no further questions or comments.

Reviewer #2 (Remarks to the Author):

I appreciate the author's detailed response and revision, my concerns are properly addressed.

Reviewer #3 (Remarks to the Author):

Thank you to the authors for their efforts in addressing my comments. The revisions have strengthened the paper, and I believe it is now suitable for publication.